# Exploring Active Learning in Meta-Learning: Enhancing Context Set Labeling

## Abstract

Most meta-learning methods assume that the (very small) context set used to establish a new task at test time is passively provided. In some settings, however, it is feasible to actively select which points to label; the potential gain from a careful choice is substantial, but the setting requires major differences from typical active learning setups. We clarify the ways in which active meta-learning can be used to label a context set, depending on which parts of the meta-learning process use active learning. Within this framework, we propose a natural algorithm based on fitting Gaussian mixtures for selecting which points to label; though simple, the algorithm also has theoretical motivation. The proposed algorithm outperforms state-of-the-art active learning methods when used with various meta-learning algorithms across several benchmark datasets.

## 1    Introduction

Meta-learning has gained significant prominence as a substitute for traditional "plain" supervised learning tasks, with the aim to adapt or generalize to new tasks given extremely limited data. (Hospedales et al. (2022) give a recent survey.) There has been enormous success compared to learning "from scratch" on each new problem, but could we do even better, with even less data?

One major way to improve data-efficiency in standard supervised learning settings is to move to an *active* learning paradigm, where typically a model can request a small number of labels from a pool of unlabeled data; these are collected, used to further train the model, and the process is repeated. (Settles (2009) provides a classic overview, and Ren et al. (2021) a more recent survey.)

Although each of these lines of research are quite developed, their combination – *active meta-learning* – has seen comparatively little research attention. Given that both focus on improving data efficiency, it seems very natural to investigate further. How can a meta-learner exploit an active learning setup to learn the best model possible, using only a very small number of labels in its context sets?

We are aware of two previous attempts at active selection of context sets in meta-learning: Müller et al. (2022) do so at meta-*training* time for text classification, while Boney & Ilin (2017) do it at meta-*test* time in semi-supervised few-shot image classification with ProtoNet (Snell et al., 2017). "Active meta-learning" thus means very different things in their procedures; these approaches are also entirely different from work on active selection of *tasks* during meta-training (as in Kaddour et al., 2020; Nikoloska & Simeone, 2022; Kumar et al., 2022). Our first contribution is therefore to clarify the different ways in which active learning can be applied to meta-learning, for differing purposes.[1]

We then confirm in extensive experiments that no active learning method for context set selection seems to significantly help with final predictor quality at meta-training time – aligning with previous observations by Setlur et al. (2020) and Ni et al. (2021) – but that active learning *can* substantially help at meta-test time. In particular, we propose a natural algorithm based on fitting a Gaussian mixture model to the unlabeled data, using meta-learned feature representations; though the approach is simple, we also give theoretical motivation. We show that our proposed selection algorithm works reliably, and often substantially outperforms competitor methods across many different meta-learning and few-shot learning tasks, across a variety of benchmark datasets and meta-learning algorithms.

---

[1] Note that work on meta-learning an active selection criterion for higher-label-budget problems – e.g. Konyushkova et al., 2017; Fang et al., 2017 – is essentially unrelated.

## 2 META-LEARNING: BACKGROUND AND WHERE TO MAKE IT ACTIVE

We aim to learn a learning algorithm $f_\theta$, a function which, given a dataset $\mathcal{C}$ consisting of pairs $(x, y) \in \mathcal{X} \times \mathcal{Y}$, returns $g := f_\theta(\mathcal{C})$. The function $g : \mathcal{X} \to \hat{\mathcal{Y}}$ is a classifier, regressor, or so on. We evaluate the quality of $g$ using a loss function $\ell : \hat{\mathcal{Y}} \times \mathcal{Y} \to \mathbb{R}$, e.g. the cross-entropy or square loss:

$$\text{\textit{Empirical risk} of } g \text{ on } \mathcal{T}: \mathcal{R}_\ell(g, \mathcal{T}) = \frac{1}{|\mathcal{T}|} \sum_{(x,y) \in \mathcal{T}} \ell\left(g(x), y\right).$$

To find the $\theta$ which gives the best $g$s, we assume we have access to distributions $\mathcal{P}^{train}, \mathcal{P}^{eval}$ over tasks $\mathcal{D} \subseteq \mathcal{X} \times \mathcal{Y}$. For each task, we will run $f_\theta$ on a *context set* $\mathcal{C}$, then evaluate the quality of the learned predictor on a disjoint *target set* $\mathcal{T}$. We call the distribution over possible $(\mathcal{C}, \mathcal{T})$ pairs $\text{Pick}_\theta(\mathcal{D})$.[2] For instance, the default choice in passive meta-learning chooses, say, five random points per class for $\mathcal{C}$ and assigns the rest to $\mathcal{T}$, ignoring $\theta$ and the $x$ values. Our aim is then roughly:

$$\text{Meta-training: find } \hat{\theta} \approx \arg\min_\theta \mathbb{E}_{\mathcal{D} \sim \mathcal{P}^{train}}\left[\mathbb{E}_{(\mathcal{C}, \mathcal{T}) \sim \text{Pick}_\theta^{train}(\mathcal{D})}\left[\mathcal{R}_{\ell^{train}}\left(f_\theta(\mathcal{C}), \mathcal{T}\right)\right]\right]. \quad (1)$$

Many algorithms have been proposed for meta-training; we give a brief overview in Section 2.2.

To compare models based on $\mathcal{P}^{eval}$, we might evaluate with a different loss. For instance, it would be typical to use the 0-1 loss (corresponding to accuracy) for classification problems.

$$\text{Meta-testing: evaluate } f_{\hat{\theta}} \text{ by estimating } \mathbb{E}_{\widetilde{\mathcal{D}} \sim \mathcal{P}^{eval}}\left[\mathbb{E}_{(\widetilde{\mathcal{C}}, \widetilde{\mathcal{T}}) \sim \text{Pick}_{\hat{\theta}}^{eval}(\widetilde{\mathcal{D}})}\left[\mathcal{R}_{\ell^{eval}}\left(f_{\hat{\theta}}(\widetilde{\mathcal{C}}), \widetilde{\mathcal{T}}\right)\right]\right]. \quad (2)$$

Finally, in practice, we might want to use a different selection scheme at deployment time. For instance, in passive meta-learning, one would typically use all available labeled data for context.

$$\text{Deployment: given a task } \check{\mathcal{D}}, \text{ find a context set via } (\check{\mathcal{C}}, \_) \sim \text{Pick}_{\hat{\theta}}^{deploy}(\check{\mathcal{D}}) \text{ and use } f_\theta(\check{\mathcal{C}}). \quad (3)$$

### 2.1 ACTIVE SELECTION OF CONTEXT IN META LEARNING

There are several places where active learning can be applied during meta-learning. In the meta-training phase (1), we could actively choose tasks $\mathcal{D}$, and/or have $\text{Pick}_\theta^{train}$ actively select points for $\mathcal{C}$ and/or $\mathcal{T}$. At meta-testing time (2), we could have $\text{Pick}_\theta^{eval}$ actively select points for $\widetilde{\mathcal{C}}$ and/or $\widetilde{\mathcal{T}}$; we might also actively choose $\widetilde{\mathcal{D}}$ to use labels efficiently, similarly to active surveying (Garnett et al., 2012). At deployment time (3), $\text{Pick}_\theta^{deploy}$ might actively choose a context set $\check{\mathcal{C}}$ to label.

Actively selecting $\mathcal{D}$, $\widetilde{\mathcal{D}}$, $\mathcal{T}$, and/or $\widetilde{\mathcal{T}}$ is interesting to minimize the label burden (or, possibly, computational cost) of meta-training (Kaddour et al., 2020; Nikoloska & Simeone, 2022; Kumar et al., 2022). We assume here, however, that $\mathcal{P}^{train}$ and $\mathcal{P}^{eval}$ are based on already-labeled datasets.

Instead, we are primarily concerned with the labeling burden at deployment time, and so would like to actively select $\check{\mathcal{C}}$ with $\text{Pick}_\theta^{deploy}$ to find the best predictor. To evaluate how well we should expect our algorithms to perform at this task, we choose $\text{Pick}_\theta^{eval} = \text{Pick}_\theta^{deploy}$; thus, we actively select $\widetilde{\mathcal{C}}$.

Should we expect this to help? Efficient approaches for data selection in meta-learning have not yet received much research attention. Setlur et al. (2020) suggest that context set diversity is empirically not particularly helpful for meta-learning, and Ni et al. (2021) show that data augmentation on context sets is not very useful either. Pezeshkpour et al. (2020) further provide some evidence using label information that there is not much room to improve few-shot classification with active learning. Agarwal et al. (2021), however, argue against the previous conclusions by showing that adversarially selected context sets, at both training and test time, significantly change the performance of few-image classification. Their approach is not applicable in practice since it requires full label information, but may suggest there is room to improve meta-learning algorithms with better context sets.

Müller et al. (2022) compare traditional active learning algorithms for few-shot text classification at training time, i.e. active $\text{Pick}_\theta^{train}$, passive $\text{Pick}_\theta^{eval}$. Boney & Ilin (2017) instead compare active

---

[2]We might want to pick points by some deterministic process, in which case $\text{Pick}_\theta(\mathcal{D})$ is a point mass.

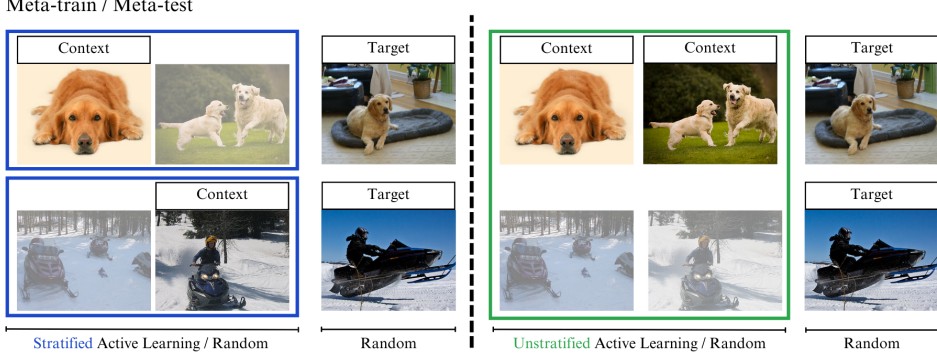

Figure 1: The meta-training process. $\text{Pick}_\theta$ can be stratified or unstratified, active or passive.

learning algorithms for semi-supervised few-shot image classification inside $\text{Pick}_\theta^{eval}$, specifically when $f_\theta$ is a ProtoNet, with passive $\text{Pick}_\theta^{train}$. Both are feasible settings, but as argued above if we are concerned with performance of our deployed predictor we should use an active $\text{Pick}_\theta^{eval} = \text{Pick}_\theta^{deploy}$. One can choose $\text{Pick}_\theta^{train}$ to be active or not, depending on which learns better predictors; we show in Appendix J that it seems active $\text{Pick}_\theta^{train}$ does not help.

**Stratification**   In passive few-shot classification, the $\text{Pick}$ functions typically choose context points according to a *stratified* sample: for one-shot classification, $\mathcal{C}$ contains exactly one point per class. This is because, if we take a uniform random sample of size $N$ for an $N$-way classification problem, $\mathcal{C}$ is unlikely to contain all the classes, making classification very difficult. Assuming "nature" gives a stratified uniform sample, as in nearly all work on few-shot classification, also seems reasonable.

In pool-based active settings, however, it is highly unreasonable to assume that $\text{Pick}_\theta^{deploy}$ can be stratified (as illustrated on the left side of Figure 1): to do so, we would need to know the label of every point in $\check{\mathcal{D}}$, in which case we should simply use all those labels. As we would like $\text{Pick}_\theta^{eval} = \text{Pick}_\theta^{deploy}$, eval-time stratification is then not particularly reasonable; even so, we report such results per the standards of meta-learning. When $\text{Pick}_\theta^{deploy}$ is unstratified (as in the right side of Figure 1), it is particularly important for the selection criterion to find samples from each class.

Train-time stratification with unstratified evaluation does not leak data labels, and is plausible when $\mathcal{P}^{train}$ and $\mathcal{P}^{eval}$ are fully labeled. Since this approach trains $f_\theta$ in an "easy" setting and evaluates it in a "hard" one, however, we will see it tends to slightly underperform the fully-unstratified default.

Regression problems are not typically stratified; we do not stratify for our regression experiments.

## 2.2   RELATED WORK: META-LEARNING ALGORITHMS

Meta-learning algorithms can be divided into several categories; all will be applicable for our active learning strategies, and we evaluate with at least one representative algorithm per category.

**Metric-based methods** learn a representation space encoding a "good" similarity, where simple classifiers work well (Vinyals et al., 2016; Finn et al., 2018; Oreshkin et al., 2018). ProtoNet (Snell et al., 2017) finds features so that points from each class are close to the prototype feature of the class.

**Optimization-based methods** use $f_\theta$ that incorporate optimization, e.g. gradient descent as in MAML (Finn et al., 2017; Antoniou et al., 2019), which seeks parameters $\theta$ (especially a parameter initialization) such that gradient descent quickly finds a useful model on a new task. ANIL (Raghu et al., 2020) freezes most of the network and only updates the last layer, while R2D2 (Bertinetto et al., 2019) and MetaOptNet (Lee et al., 2019b) replace the last layer with a convex problem whose solution can be differentiated; these approaches can improve both performance and speed.

**Model-based methods** learn a model that explicitly adapts to new tasks, typically by modeling the distribution of $y$ from $\mathcal{T}$ given its $x$ values and $\mathcal{C}$. The most prominent family of methods is Neural Processes (NPs) (Garnelo et al., 2018b; Dubois et al., 2020), which encode a context set and estimate

task-specific distribution parameters. Conditional (and latent) NPs can have issues with underfitting (Garnelo et al., 2018a;b), but AttentiveNPs (Kim et al., 2019) and ConvNPs (Gordon et al., 2020) can be more powerful. These models are more commonly used for regression.

**Pre-training methods**, such as SimpleShot (Wang et al., 2019) and Baseline++ (Chen et al., 2019), are based on repeated demonstrations (also Zhang et al., 2021; Goldblum et al., 2020) that simply pre-training a multi-class model can surpass the performance of commonly used meta-learners.

## 3 ACTIVE LEARNING

In pool-based active learning, a model requests labels for the most "informative" data points from a pool of unlabeled data. The key question is how to estimate which data points will be informative.

### 3.1 RELATED WORK: EXISTING ACTIVE LEARNING METHODS

**Uncertainty-based methods** Simple but effective uncertainty-based methods such as maximum entropy (Wang & Shang, 2014), least confident (Settles, 2009), and margin sampling (Scheffer et al., 2001) are widely used for active learning. Since they only consider current models' uncertainty, active learning strategies that consider expected changes in model parameters (Settles et al., 2007; Ash et al., 2020) and model outputs (Zhu et al., 2003; Guo & Greiner, 2007; Roy & McCallum, 2001; Käding et al., 2018; Freytag et al., 2014; Käding et al., 2016; Tan et al., 2021; Mohamadi et al., 2022) have been also been proposed. However, recent analyses have empirically demonstrated that at least in certain experimental settings, most active learning methods are not significantly different from one another (Lang et al., 2021), and may not even improve over random selection (Munjal et al., 2022).

**Random** Uniformly randomly samples a context set from all the candidate data points.

**Entropy** Add one point to the context set based on $x^* = \arg\max_{x \in \mathcal{U}} H(\hat{y}(x) \mid x)$, where $\mathcal{U}$ are the unlabeled candidate data points and $H(\cdot)$ is Shannon entropy (Wang & Shang, 2014). Other than in Appendix K, we apply this in "batch mode," i.e. we do not observe points one-by-one but rather choose the $|\mathcal{C}|$ points with the highest "initial" entropy.[3]

**Margin** For classification, add one point to $\mathcal{C}$ based on $x^* = \arg\min_{x \in \mathcal{U}} p_1(y|x) - p_2(y|x)$, where $p_1$ and $p_2$ denote the first and second highest predicted probabilities, respectively (Scheffer et al., 2001). We also run this method in "batch mode."

Although Entropy and Margin are very simple and fast to evaluate, no uncertainty-based method seems to substantially outperform them on typical active image classification tasks (e.g. Mohamadi et al., 2022), and we will see that other methods are unlikely to be competitive in low-budget regimes.

**Low-budget active learning** The limitations of typical active learning approaches may especially apply in very-low-budget cases, such as those considered in few-shot classification and meta-learning. In particular, when the "current" model is quite bad, using it to choose points might be counterproductive. In the one-shot case especially, standard active learning methods simply do not apply.

Recently, several papers have have proposed novel active learning algorithms for these settings; none of these papers focused on meta-learning, but should be broadly applicable since meta-learning is also a low-budget setting. Rather than picking e.g. the points about which a model is least certain, these papers propose to label the "most representative" data points independently of a "current" model.

**DPP** Determinantal Point Processes (DPPs) can query diverse samples, based on selecting a subset that maximizes the determinant of a similarity matrix (Bıyık et al., 2019).

**Typiclust** Run $k$-means on the unlabeled data points, where $k = |\mathcal{C}|$ is the annotation budget. Select one data point per cluster such that the distance between a data point and its $k'$ nearest neighbors is minimized: $\arg\min_{x \in \mathcal{U}} \sum_{x' \in \text{NN}_{k'}(x)} \|x - x'\|_2$ (Hacohen et al., 2022).

---

[3]Traditional active learning methods would generally retrain between each step, requiring a back-and-forth labeling process not needed by the methods discussed shortly. In modern deep learning settings, this is almost never done due to the expense of retraining; "batch-mode" entropy is still excellent in those settings (Lang et al., 2021; Mohamadi et al., 2022). Appendix K explores more frequent retraining; the takeaway results are overall similar to the rest of our experiments.

**Coreset** Greedily select a subset of the unlabeled data points to approximately minimize the distance from unlabeled data points to their nearest labeled point (Sener & Savarese, 2018).

**ProbCover** Select data points that roughly maximize the number of unlabeled points within a distance of $\tau$ from any labeled point, where $\tau$ is chosen according to a "purity" heuristic (Yehuda et al., 2022); see Appendix F for more details.

## 3.2 Features for Representative-Selection Methods

Notions of the "most representative" data points are highly dependent on a reasonable metric of data similarity. Prior methods operated either on raw data – typically a poor choice for complex datasets like natural images – or, in semi-supervised settings as in ProbCover and Typiclust, on SimCLR (Chen et al., 2020) features learned on the unlabeled data.

In metric-based meta-learning, we propose to instead use the current meta-learned representation; choosing points representative for the features we will use downstream is the natural choice.

In MAML, the most natural equivalent might be features from the empirical neural tangent kernel (Lee et al., 2019a) of the current initialization network; this approximates what will happen when the network is trained on $\mathcal{C}$,[4] and so is perhaps the best simple understanding of "how this network views the data." Even empirical NTKs are often expensive to evaluate, however, and we thus propose to instead use features from the penultimate layer of the initialization neural net $f_\theta(\{\})$, corresponding to the NTK of a model that only retrains its last layer (as in ANIL, R2D2, and MetaOptNet).

We also use the penultimate-layer reperesentations of $f_\theta(\{\})$ for NP-based meta-learning.

Experiments in Appendix I show that this proposal outperforms separate self-supervised features.

## 3.3 Gaussian Mixture Selection for Low-Budget Active Learning

We propose the following very simple algorithm for low-budget active learning: fit a mixture of $k$ Gaussians to the unlabeled data features, where $k$ is the label budget, using EM with a $k$-means initialization. We use a shared diagonal covariance matrix. Once a mixture is fit, we select the highest-density point from each component: $\arg\min_{x\in\mathcal{U}}(x-\mu_j)^\mathsf{T}\Sigma^{-1}(x-\mu_j)$ for each $j \in [k]$.

For metric-based meta-learning, the motivation of this algorithm is clear: we want labeled points that approximately "cover" the data points. Our notion of a "cover" is somewhat different from that of Coreset (Sener & Savarese, 2018) or ProbCover (Yehuda et al., 2022); we avoid ProbCover's need for a fixed radius, which we show can lead to poor choices (see Appendix F), and are more concerned with "average" covering (and hence perhaps less sensitive to outliers) than Coreset. The quality of selected data points from those methods are compared according to a few metrics in Figure 6.

On ANIL and MetaOptNet: since $|\mathcal{C}|$ is at most, say, 50 (in 10-way 5-shot) and the feature dimension is typically hundreds, ANIL becomes approximately the same multiclass max-margin separator obtained by (unregularized) MetaOptNet.[5] Intuitively, as $|\mathcal{C}|$ grows, the means of an isotropic Gaussian mixture converge to roughly a covering set for the dataset $\mathcal{U}$, and the max-margin separator of a set cover for $\mathcal{U}$ will be similar to the max-margin separator for all of the data. Even in various cases when $|\mathcal{C}| \ll |\mathcal{U}|$, choosing the means yields a max-margin separator that generalizes well.

Figure 4 in Appendix A illustrates that, if class-conditional data distributions are isotropic Gaussians with the same covariance matrices, labeling the cluster centers can be far preferable to labeling a random point from each cluster. This is backed up by the following result in a particular simple case:

**Proposition 1.** *Suppose $Y \sim \text{Uniform}([N])$, and $X \mid (Y = y) \sim \mathcal{N}(\mu_y, \sigma^2 I)$, where the $\mu_i$ are orthonormal. Then the max-margin separator (4) on $\{(\mu_i, i)\}_{i=1}^N$ is Bayes-optimal for $Y \mid (X = x)$.*

For more general settings, we argue that GMM is still a good method based on being an efficient set cover, as shown in Figure 5 in Appendix A along with the proof for Proposition 1.

---

[4]Theoretical results about the NTK technically depend on a random initialization, which is not the case here. Mohamadi et al. (2022) provide some assurance in that if the initialization were obtained by gradient descent on some dataset, the results would still hold, but MAML finds initial parameters differently.

[5]For reasonable distributions and networks, $\mathcal{C}$ is almost surely linear separable; thus ANIL, which is gradient descent for logistic regression, will converge to the multiclass max-margin separator (Soudry et al., 2018).

**Very-low-budget regime**   Active learning based on Gaussian mixtures is not new in itself. Closely-related methods such as $k$-means, $k$-means$^{++}$ or $k$-medoids have been used as sole selection algorithms (Aghaee et al., 2016; Voevodski et al., 2012) or in combination with uncertainty-based methods (Nguyen & Smeulders, 2004; Donmez et al., 2007; Ash et al., 2020; Hacohen et al., 2022). Some recent work (Boney & Ilin, 2017) including DPP (Bıyık et al., 2019) and Coreset (Sener & Savarese, 2018) show significant improvements over GMM-based baselines. These trends, however, do not seem to hold true in very-low-budget regimes such as meta-learning. As shown in Figure 2, GMM matches or outperforms other low-budget methods with very small numbers of labels for standard image classification tasks; the following section shows that GMM provides substantial improvements in meta-learning.

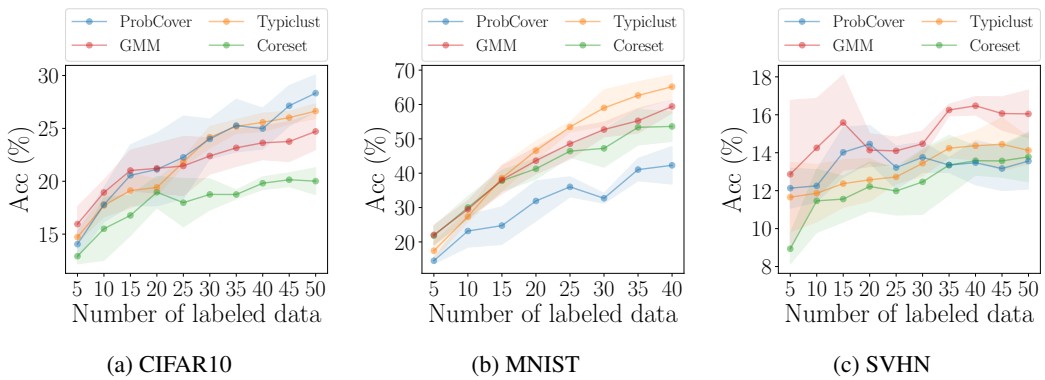

(a) CIFAR10        (b) MNIST        (c) SVHN

Figure 2: Low-budget active learning methods on image classification with very low budget. Mean and standard error of accuracy for three sets of SimCLR features, three runs per features.

## 4    ACTIVE META-LEARNING EXPERIMENTS

We now compare various active learning methods for variants of active meta learning as defined in Section 2.1, on both classification tasks (in Sections 4.1 and 4.2) and regression (in Section 4.3).

### 4.1    FEW-SHOT IMAGE CLASSIFICATION

We use four popular few-shot image classification benchmark datasets. **MiniImageNet** (Vinyals et al., 2016; Ravi & Larochelle, 2017) consists of 60 000 images sampled from ImageNet (Deng et al., 2009), with 64 train classes, 16 validation, and 20 test. Each class has 600 images. **TieredImageNet** (Ren et al., 2018) is a much larger subset of ImageNet; it consists of 34 super-classes, each containing 10 to 30 sub-classes. There are 20 train super-classes, 6 validation, and 8 test. **FC100** (Oreshkin et al., 2018) is a subset of CIFAR100 (Krizhevsky, 2009), with 60 train classes, 20 validation, and 20 test. It is designed to minimize the overlap between the splits. **CUB** (Wah et al., 2011; Hilliard et al., 2018) consists of 200 classes of bird images, with 140 train classes, 30 validation, and 30 test.

We validate whether our active learning methods work across various types of meta-learning algorithms. We run[6] metric-based: ProtoNet (Snell et al., 2017), optimization-based: MAML (Finn et al., 2017), ANIL (Raghu et al., 2020), and MetaOpt(Lee et al., 2019b), as well as pre-training-based: Baseline++ (Chen et al., 2019) and SimpleShot (Wang et al., 2019).[7]We vary the backbone to demonstrate robustness: for instance, we use 4 convolutional blocks for MAML and ProtoNet, and ResNet10 (He et al., 2016) for Baseline++. As typical in few-shot classification, we report means and 95% confidence intervals for test accuracy from a single model, based on 600 meta-test samples.

We use the meta-learner's features as proposed in Section 3.2 for all methods; experiments in Appendix I confirm that they outperform contrastive learning of features on the meta-training set.

---

[6]We reproduce ProtoNet, MAML, and ANIL models using the Learn2Learn library (Arnold et al., 2020); for MetaOptNet, Baseline++, and SimpleShot, we used the original repositories provided by the authors.

[7]We do not run a model-based method on this case, though we will in Section 4.3; most variants do not work well conditioning on images, and the ones that do are far less commonly applied in these contexts.

| $\text{Pick}_\theta^{eval}$ | 1-Shot | | | 5-Shot | | |
|---|---|---|---|---|---|---|
| | Fully strat. | Train strat. | Unstrat. | Fully strat. | Train strat. | Unstrat. |
| Random | $36.73 \pm 0.18$ | $31.27 \pm 0.21$ | $31.40 \pm 0.41$ | $47.98 \pm 0.18$ | $42.83 \pm 0.20$ | $44.00 \pm 0.21$ |
| Entropy | $33.67 \pm 0.16$ | $29.82 \pm 0.20$ | $30.01 \pm 0.20$ | $44.64 \pm 0.17$ | $38.39 \pm 0.22$ | $38.36 \pm 0.25$ |
| Margin | $34.28 \pm 0.18$ | $29.74 \pm 0.20$ | $28.99 \pm 0.20$ | $45.31 \pm 0.17$ | $39.65 \pm 0.21$ | $38.13 \pm 0.24$ |
| DPP | $36.20 \pm 0.18$ | $31.34 \pm 0.20$ | $31.09 \pm 0.20$ | $47.53 \pm 0.17$ | $43.69 \pm 0.20$ | $44.19 \pm 0.20$ |
| Coreset | $35.79 \pm 0.17$ | $30.31 \pm 0.20$ | $31.57 \pm 0.18$ | $43.08 \pm 0.40$ | $41.56 \pm 0.20$ | $41.79 \pm 0.22$ |
| Typiclust | $46.01 \pm 0.16$ | $30.96 \pm 0.19$ | $30.61 \pm 0.21$ | $47.54 \pm 0.17$ | $43.61 \pm 0.18$ | $44.03 \pm 0.21$ |
| ProbCover | $48.66 \pm 0.16$ | $32.86 \pm 0.22$ | $33.58 \pm 0.19$ | $51.11 \pm 0.17$ | $44.20 \pm 0.23$ | $44.40 \pm 0.24$ |
| GMM (Ours) | $50.22 \pm 0.18$ | $34.23 \pm 0.23$ | $35.03 \pm 0.23$ | $54.76 \pm 0.17$ | $46.30 \pm 0.21$ | $47.03 \pm 0.20$ |

Table 1: 5-Way K-Shot classification on FC100 with ProtoNet, with $\text{Pick}_\theta^{train}$ random. The **first**, **second**, **third** best results for each setting are marked in this and all other results tables.

| $\text{Pick}_\theta^{eval}$ | 1-Shot | | | 5-Shot | | |
|---|---|---|---|---|---|---|
| | Fully strat. | Train strat. | Unstrat. | Fully strat. | Train strat. | Unstrat. |
| Random | $47.93 \pm 0.20$ | $28.16 \pm 0.17$ | $34.85 \pm 0.19$ | $64.16 \pm 0.18$ | $53.54 \pm 0.20$ | $58.84 \pm 0.20$ |
| Entropy | $48.16 \pm 0.20$ | $25.56 \pm 0.14$ | $30.44 \pm 0.17$ | $61.22 \pm 0.20$ | $34.36 \pm 0.23$ | $39.57 \pm 0.26$ |
| Margin | $48.31 \pm 0.20$ | $28.32 \pm 0.16$ | $30.83 \pm 0.17$ | $63.73 \pm 0.18$ | $49.24 \pm 0.22$ | $53.92 \pm 0.22$ |
| DPP | $48.96 \pm 0.21$ | $28.90 \pm 0.17$ | $36.44 \pm 0.19$ | $64.15 \pm 0.18$ | $54.18 \pm 0.20$ | $57.86 \pm 0.19$ |
| Coreset | $47.74 \pm 0.20$ | $29.19 \pm 0.18$ | $33.71 \pm 0.18$ | $61.28 \pm 0.18$ | $30.98 \pm 0.19$ | $45.74 \pm 0.23$ |
| Typiclust | $55.65 \pm 0.18$ | $27.45 \pm 0.17$ | $35.46 \pm 0.18$ | $64.16 \pm 0.18$ | $46.70 \pm 0.21$ | $57.83 \pm 0.21$ |
| ProbCover | $52.07 \pm 0.17$ | $23.34 \pm 0.11$ | $37.29 \pm 0.18$ | $64.66 \pm 0.18$ | $40.01 \pm 0.21$ | $45.32 \pm 0.22$ |
| GMM (Ours) | $58.82 \pm 0.24$ | $33.34 \pm 0.24$ | $37.68 \pm 0.19$ | $67.18 \pm 0.18$ | $54.35 \pm 0.20$ | $59.05 \pm 0.20$ |

Table 2: 5-Way K-Shot classification on MiniImageNet with MAML, with $\text{Pick}_\theta^{train}$ random.

Additionally, in the main body we only present results where $\text{Pick}_\theta^{train}$ is random; Appendix J demonstrates that, in our setup, active learning at train time is actually mildly *harmful* to overall performance, which aligns with the observations in Ni et al. (2021) and Setlur et al. (2020).

For **metric-based** methods, Table 1 shows results for ProtoNet on FC100. The simple GMM method significantly outperforms the other active learning strategies on all problem variants considered here. As previously reported (Hacohen et al., 2022; Yehuda et al., 2022), uncertainty-based methods are significantly worse than random selection in this low-budget regime.

For **optimization-based**, Table 2 shows results with MAML on MiniImageNet. Similar to Table 1, GMM again significantly outperforms the other strategies in most cases. The performance of ProbCover is sometimes much lower than other methods due to its radius parameter, which is very difficult to tune, with the best choice changing dramatically depending on the sub-task even though Yehuda et al. (2022) proposed to fix this parameter per dataset (see Appendix F for more). Results for ANIL on TieredImageNet and MetaOptNet on FC100 are provided in Appendix G.

For **pre-training-based** methods, we compare active learning strategies with Baseline++ on the CUB dataset in Table 3, seeing that the proposed method is again usually by far the best, though in one five-shot case it essentially ties DPP. As these methods do not follow the meta-training process in (1), train-time stratification is not applicable. Appendix G shows results for SimpleShot.

**Comparison between active learning methods.** Figure 3 (left) visualizes context set selection using t-SNE (van der Maaten & Hinton, 2008) for one 5-way, 1-shot, unstratified task. It is vital to select one sample from each class; only GMM does so here. Figure 3 (right) summarizes behavior across many tasks; while not perfect, GMM does a much better job of selecting distinct classes.

**Entropy** and **Margin** are typically far worse than random. So is **Coreset**, agreeing with prior observations (Ash et al., 2020; Hacohen et al., 2022; Yehuda et al., 2022); this may be because of issues with the greedy algorithm and/or sensitivity to outliers. **Typiclust** tends to pick points which, while dense according to its "typicality measure," are far from cluster centers; this may be helpful in traditional active learning, but seems to hurt here. **DPP** is often better than random, but only barely; diverse selections may not necessarily lead to representative selections.

| $\mathrm{Pick}_\theta^{eval}$ | 1-Shot | | 5-Shot | |
|---|---|---|---|---|
| | Test strat. | Test unstrat. | Test strat. | Test unstrat. |
| Random | $68.44 \pm 0.92$ | $51.03 \pm 0.88$ | $82.66 \pm 0.56$ | $\textbf{79.57} \pm \textbf{0.67}$ |
| Entropy | $66.33 \pm 0.91$ | $45.31 \pm 0.89$ | $80.97 \pm 0.60$ | $78.33 \pm 0.72$ |
| Margin | $68.65 \pm 0.90$ | $50.48 \pm 0.94$ | $\textbf{82.29} \pm \textbf{0.64}$ | $71.07 \pm 0.83$ |
| DPP | $\textbf{71.53} \pm \textbf{0.89}$ | $54.38 \pm 0.92$ | $\textbf{82.81} \pm \textbf{0.55}$ | $\textbf{78.62} \pm \textbf{0.76}$ |
| Coreset | $69.01 \pm 0.91$ | $\textbf{56.22} \pm \textbf{0.94}$ | $82.07 \pm 0.55$ | $76.35 \pm 0.74$ |
| Typiclust | $70.58 \pm 0.81$ | $29.80 \pm 0.32$ | $74.86 \pm 0.81$ | $70.00 \pm 0.92$ |
| ProbCover | $\textbf{78.11} \pm \textbf{0.69}$ | $\textbf{55.09} \pm \textbf{0.98}$ | $78.59 \pm 0.64$ | $65.71 \pm 0.97$ |
| GMM (Ours) | $\textbf{79.98} \pm \textbf{0.60}$ | $\textbf{59.55} \pm \textbf{0.87}$ | $\textbf{82.55} \pm \textbf{0.58}$ | $\textbf{82.68} \pm \textbf{0.57}$ |

Table 3: 5-Way K-Shot classification on CUB with Baseline++, with $\mathrm{Pick}_\theta^{train}$ random.

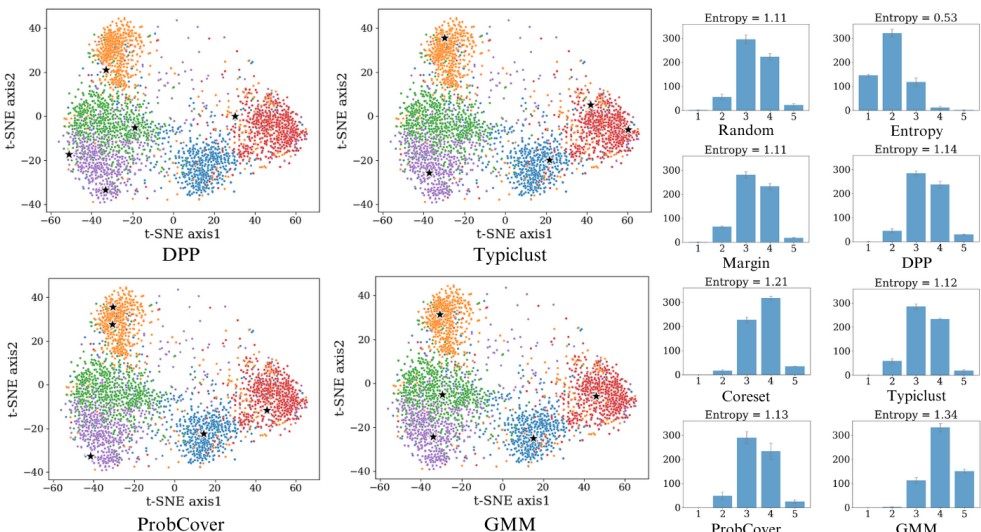

Figure 3: **Left.** t-SNE visualization of unlabeled points of one 5-way, 1-shot, unstratified MiniImageNet task. Stars denote selected context points using each method. **Right.** Distributions of the number of classes selected in each $\widetilde{\mathcal{C}}$ by ProtoNet on MiniImageNet among 600 meta-test cases, along with the mean empirical entropy of $y$ from $\widetilde{\mathcal{C}}$. The higher the value is, the more diverse classes an active learning method selects; $\log 5 \approx 1.6$ would be perfect.

**ProbCover** manages to cover the feature space well, and is usually second-best. However, its "hard" radius causes issues; it may be preferable to use a smoother notion, as in GMM. The "purity" heuristic to choose a radius $\delta$ also does not seem to align well with performance for meta-learning, as shown in Appendix F. More analysis for the poor performance of other methods are provided in Appendix N.

**GMM**, by contrast, provides robust performance without introducing significant new hyperpameters.[8]

"Soft" $k$-means would be a special case of GMM with a spherical covariance. For some cases, standard $k$-means performs about the same as the GMM, but the GMM is occasionally much better: for Baseline++ on CUB, GMM outperforms $k$-means by $3.95$ points for 5-way 1-shot and $11.79$ for 5-shot. We provide a more thorough comparison to $k$-means in Appendix E.

## 4.2 CROSS-DOMAIN ACTIVE META-LEARNING

Cross-domain learning, where $\mathcal{P}^{train}$ is "fundamentally different" from $\mathcal{P}^{eval}$, is typically more difficult than "in-domain" meta-learning. We use a ResNet18 (He et al., 2016) pretrained with standard supervised learning on ImageNet, and meta-test on CUB and **Places** (Zhou et al., 2017),

---

[8]We did not significantly tune the $k$-means or EM optimization parameters from standard defaults.

| $\text{Pick}_\theta^{eval}$ | $P^{eval}$ on Places | | $P^{eval}$ on CUB | |
|---|---|---|---|---|
| | 1-Shot | 5-Shot | 1-Shot | 5-Shot |
| Random | $44.28 \pm 1.93$ | $77.92 \pm 1.70$ | $49.93 \pm 0.92$ | **$84.38 \pm 0.72$** |
| Entropy | $36.12 \pm 1.25$ | $57.79 \pm 2.93$ | $41.85 \pm 0.99$ | $71.15 \pm 0.99$ |
| Margin | $43.31 \pm 1.97$ | $73.65 \pm 1.94$ | $48.04 \pm 0.98$ | $78.84 \pm 0.92$ |
| DPP | $46.76 \pm 2.29$ | **$78.36 \pm 1.89$** | **$51.41 \pm 0.90$** | $84.19 \pm 0.72$ |
| Coreset | **$50.03 \pm 0.93$** | $65.20 \pm 2.77$ | $50.77 \pm 0.95$ | $81.80 \pm 0.81$ |
| Typiclust | $43.76 \pm 1.98$ | **$77.57 \pm 1.84$** | $43.39 \pm 1.03$ | $50.69 \pm 1.08$ |
| ProbCover | **$47.93 \pm 1.08$** | $59.08 \pm 2.50$ | **$62.13 \pm 1.08$** | $69.80 \pm 1.16$ |
| GMM (Ours) | **$60.01 \pm 0.86$** | **$86.45 \pm 1.42$** | **$59.87 \pm 0.86$** | **$85.49 \pm 0.67$** |

Table 4: Cross-domain meta-learning tasks using a ResNet18 pre-trained on ImageNet.

| Active Strategy | Sine func. (3-Shots) | Distractor (2-Shots) | | ShapeNet1D (2-Shots) | |
|---|---|---|---|---|---|
| | | IC | CC | IC | CC |
| Random | $24.17 \pm 0.43$ | $18.91 \pm 2.13$ | $25.79 \pm 2.17$ | $16.52 \pm 1.08$ | $19.07 \pm 1.30$ |
| DPP | **$23.19 \pm 0.51$** | **$18.08 \pm 2.12$** | **$19.68 \pm 1.92$** | **$11.83 \pm 0.85$** | **$13.68 \pm 0.93$** |
| Coreset | $31.36 \pm 0.48$ | **$19.58 \pm 1.95$** | **$24.08 \pm 2.19$** | **$11.39 \pm 0.91$** | **$13.05 \pm 1.18$** |
| Typiclust | **$21.59 \pm 0.40$** | $20.27 \pm 2.15$ | $24.96 \pm 2.68$ | $12.54 \pm 1.08$ | $14.58 \pm 1.24$ |
| ProbCover | $29.36 \pm 0.49$ | $21.96 \pm 2.45$ | $25.25 \pm 2.78$ | $12.31 \pm 0.85$ | $13.95 \pm 1.08$ |
| GMM (Ours) | **$18.09 \pm 0.38$** | **$17.95 \pm 2.05$** | **$22.03 \pm 2.42$** | **$10.78 \pm 0.72$** | **$12.35 \pm 0.97$** |

Table 5: Meta-learning for regression on a toy dataset and two pose estimation datasets for Intra-Category (IC) and Cross-Category (CC). Sine func. and Distractor use mean squared error, ShapeNet1D uses cosine-sine-distance; lower values are better for each.

which contains images of "places" such as restaurants. As used for cross-domain meta-learning by Oh et al. (2022), it contains 16 classes with an average of 1,715 images each. As the model is not meta-trained, train stratification is not relevant; we show results in Table 4 only for unstratified test sets. GMM is again the clear overall winner; all other methods are often worse than random.

### 4.3 ACTIVE META-LEARNING FOR REGRESSION

Each **sinusoidal function** (Finn et al., 2017) has task $y = a \sin(x + p)$, where $a \sim \text{Unif}(0.1, 5)$ is the amplitude, and $p \sim \text{Unif}(0, \pi)$ is the phase of sine functions; we use MAML for this dataset.

**Distractor** and **ShapeNet1D** are vision regression datasets (Gao et al., 2022); the task is to predict the position of a specific object in an image ignoring a distractor, or to predict an object's 1D pose (azimuth rotation). **IC** uses objects whose classes were observed during meta-training, while **CC** has novel object classes. We use conditional Neural Processes (NP) for Distractor, and attentive NP for ShapeNet1D. Details are provided in Appendix H.

Table 5 compares active strategies on these datasets; GMM again performs generally the best.

## 5 DISCUSSION

We have clarified the ways in which active learning can be incorporated into meta-learning. While active context set selection does not seem to work at meta-training time (Appendix J), it can be extremely useful at meta-testing/deployment time.

We proposed a surprisingly simple method that substantially outperforms previous proposals. It is simple and intuitive, and bears some theoretical guarantees in a particular simple situation, though why it improves so thoroughly over related methods is not yet fully clear.

While we evaluated on a range of methods across many datasets, we focused on convolutional models for computer vision tasks; although we see no particular reason to expect this, it's conceivable that things might behave differently with other types of data and/or models.

**Reproducibility** For each experiment, we listed implementation details of the experiment such as model, optimizer, batch size, and training iterations, along with the links to publicly available code we built on, in Appendix B. Anonymous code is provided in the supplementary material. For the theoretical result, we clearly mentioned conditions and provided a complete proof in Appendix A.

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

# A    DETAILS FOR MAX-MARGIN MOTIVATION

The following optimization problem is one form of an $N$-class max-margin problem, i.e. a multi-class support vector machine (Crammer & Singer, 2001), on a training set $\{(x_i, y_i)\}_{i=1}^{m}$:

$$\min_{w_1,\ldots,w_N} \sum_{y=1}^{N} \|w_y\|^2 \quad \text{s.t.} \quad \forall i \in [m], \; \forall y' \neq y_i, \; w_{y_i}^\mathsf{T} x_i \geq w_{y'}^\mathsf{T} x_i + 1. \tag{4}$$

This is a "hard" version of the problem used as a classification head by MetaOptNet (Lee et al., 2019b), and can be obtained in their framework by taking the penalty parameter $C \to \infty$.

and can be obtained in their framework by taking the penalty parameter $C \to \infty$.

The decision boundaries obtained by small-step-size gradient descent for linear predictors with cross-entropy loss on separable data converge to those obtained by (4), as shown by Soudry et al. (2018, Theorem 7), for almost all datasets. Thus, ANIL (Raghu et al., 2020), which uses gradient descent for linear predictors with cross-entropy loss on separable data, will approximately obtain the same solution when using enough steps with appropriately small learning rates.

MetaOptNet uses the homogeneous predictors discussed here. We can handle non-homogeneous linear predictors ($w^\mathsf{T} x + b$ instead of just $w^\mathsf{T} x$) with the standard trick of adding a constant 1 feature to each data point. This solution actually does not quite maximize the margin on the original problem, since it effectively adds $b^2$ to the objective in (4), but ANIL will find exactly this same solution when using gradient descent on a function with a separate intercept term.

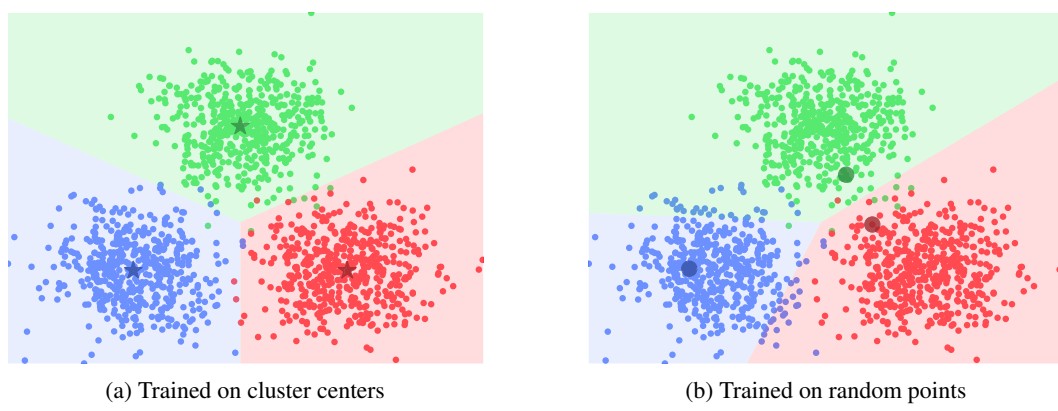

(a) Trained on cluster centers    (b) Trained on random points

Figure 4: Decision boundaries using a multiclass SVM (4) trained on a one-shot dataset containing (a) cluster centers (shown by stars) and (b) randomly selected points (shown by circles).

Figure 4 demonstrates visually that, if the class-conditional data distributions are isotropic Gaussians with the same covariance matrices, labeling the cluster centers can be far preferable to labeling a random point from each cluster. This is backed up by the following result in a particular case:

**Proposition 1.** *Suppose $Y \sim \text{Uniform}([N])$, and $X \mid (Y = y) \sim \mathcal{N}(\mu_y, \sigma^2 I)$, where the $\mu_i$ are orthonormal. Then the max-margin separator (4) on $\{(\mu_i, i)\}_{i=1}^{N}$ is Bayes-optimal for $Y \mid (X = x)$.*

*Proof.* Combine Lemmas 1 and 2 below. □

The orthonormal assumption keeps the proof tractable; far more analysis would be needed without it. With high-dimensional meta-learned features that are well-aligned to the learning problem, however, it is reasonable to expect that inner products between different classes will be much smaller than the within-class inner products.

This optimality result can break when the clusters do not share a spherical covariance; consider Figure 5a, where the data is still Gaussian but the shared class-conditional covariance is not spherical. In the one-shot case, max-margin on the separators does not choose the optimal separator. In this case, we could manually select points to choose the correct line. Doing so, however, is quite risky;

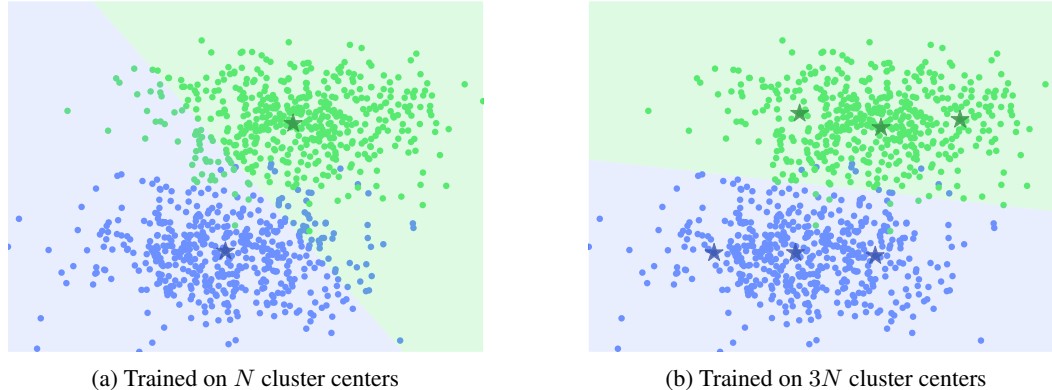

(a) Trained on $N$ cluster centers          (b) Trained on $3N$ cluster centers

Figure 5: Decision boundaries using a multiclass SVM (4) trained on cluster centers (shown by stars), with (a) the one-shot case and (b) the three-shot case.

since we do not know the data labels (or that it is actually Gaussian), we might incorrectly separate the data. Figure 5b shows the same problem in a three-shot setting; here, even though the data is truly generated from a mixture of two Gaussians, fitting a mixture of six Gaussians gives us an approximate set cover of the data, and the max-margin separator now works well.

In fact, we can expect that (a) as the number of clusters grows, the cluster centers produce a better and better set cover of the dataset; (b) the max-margin separator on a set cover will approximate the max-margin separator on the full dataset, since the support vectors are all nearby.

## A.1 Proofs

**Lemma 1.** *Suppose that $\{x_i\}_{i=1}^N$ are orthonormal. Then the solution to (4) with the dataset $\{(x_y, y)\}_{y=1}^N$ is given by $w_y = x_y - \frac{1}{N}\sum_{i=1}^N x_i$, and hence*

$$\arg\max_y w_y^\mathsf{T} x = \arg\min_y \|x - x_y\|.$$

*Proof.* We will be able to analytically solve the KKT conditions for (4) in this case. Rather than using existing analyses of (4), it will be simpler to directly analyze this particular case.

Let $\mathbf{w} = \begin{bmatrix} w_1 \\ \vdots \\ w_N \end{bmatrix} \in \mathbb{R}^{Nd}$, where $d$ is the dimension of the $x_i$ and $w_y$. The objective of our optimization problem is then simply $\|\mathbf{w}\|^2$.

We will next define a matrix $A$ such that the constraints can be written as $A\mathbf{w} + \mathbf{1} \leq \mathbf{0}$, with $A \in \mathbb{R}^{N(N-1)\times Nd}$ and $\leq$ interpreted elementwise. Each constraint is of the form $-w_i^\mathsf{T} x_i + w_j^\mathsf{T} x_i + 1 \leq 0$, where $i \neq j$ are class indices in $[N]$. We can write the corresponding row of $A$ as $(E_j - E_i)x_i$, where $E_i \in \mathbb{R}^{Nd\times d}$ are given by $E_i = \begin{bmatrix} 0_{(i-1)d\times d} \\ I_d \\ 0_{(N-i-1)d\times d} \end{bmatrix}$; these $E_i$ are a block-matrix analogue of standard basis vectors, so that $E_i x_i \in \mathbb{R}^{Nd}$ has $x_i$ in the $i$th block of $d$ coordinates, and 0 elsewhere. We will order these constraints in $A$ in "row-major" order: recalling that $i \neq j$, this means we have first $i = 1$ $j = 2$, then $i = 1$ $j = 3$, up to $i = 1$ $j = N$, followed by $i = 2$ $j = 1$, $i = 2$ $j = 3$, and so on. Let $\ell(i, j)$ give the index of the corresponding constraint, so that e.g. $\ell(1, 3) = 2$.

Now, the problem can be written

$$\min_{\mathbf{w}\in\mathbb{R}^{Nd}} \frac{1}{2}\|\mathbf{w}\|^2 \text{ s.t. } A\mathbf{w} + \mathbf{1} \leq \mathbf{0},$$

with the $\frac{1}{2}$ introduced for convenience. The KKT conditions for this problem are

$$\mathbf{w} + A^\mathsf{T}\mu = \mathbf{0} \quad A\mathbf{w} + \mathbf{1} \leq \mathbf{0} \quad \mu \geq \mathbf{0} \quad \mu \odot (A\mathbf{w} + \mathbf{1}) = \mathbf{0},$$

where $\odot$ is elementwise multiplication. From the first condition, $\mathbf{w} = -A^\mathsf{T}\mu$, where $\mu \in \mathbb{R}^{N(N-1)}$ is any vector satisfying

$$\mu \geq \mathbf{0} \quad AA^\mathsf{T}\mu - \mathbf{1} \geq \mathbf{0} \quad \mu \odot (AA^\mathsf{T}\mu - \mathbf{1}) = \mathbf{0}.$$

Since (4) is a strictly convex minimization problem with affine constraints, these conditions are necessary and sufficient for optimality, and the solution $\mathbf{w}$ is unique.

We can reasonably expect, since the $x_i$ are orthonormal, that all constraints should be active, meaning that $AA^\mathsf{T}\mu = \mathbf{1}$. Indeed, choosing $\mu = (AA^\mathsf{T})^{-1}\mathbf{1}$ automatically satisfies the second and third conditions; it only remains to show that this $\mu \geq \mathbf{0}$ in order to show this as an optimal solution to (4).

To do this, we will explicitly characterize $AA^\mathsf{T}$:

$$(AA^\mathsf{T})_{\ell(i,j),\ell(i',j')} = x_i^\mathsf{T}(E_j - E_i)^\mathsf{T}(E_{j'} - E_{i'})x_{i'} = (\delta_{ii'} + \delta_{jj'} - \delta_{ij'} - \delta_{ji'})\, x_i^\mathsf{T} x_{i'},$$

where $\delta_{ij} = \mathbb{1}(i = j)$ is the Kronecker delta, since $E_i^\mathsf{T} E_j = \delta_{ij} I_d$.

Since the $x_i$ are orthonormal, $x_i^\mathsf{T} x_{i'} = \delta_{ii'}$. As we know $i \neq j$ and $i' \neq j'$, this simplifies to

$$(AA^\mathsf{T})_{\ell(i,j),\ell(i',j')} = \delta_{ii'}(1 + \delta_{jj'}).$$

Thus $(AA^\mathsf{T})$ is a block matrix with diagonal blocks of size $(N-1) \times (N-1)$ with values $I_{N-1} + \mathbf{1}_{N-1}\mathbf{1}_{N-1}^\mathsf{T}$, and all off-diagonal blocks zero. Taking $\mu = (AA^\mathsf{T})^{-1}\mathbf{1}_{N(N-1)}$, the zero blocks contribute nothing, so each block of $N-1$ entries of $\mu$ is $(I_{N-1} + \mathbf{1}_{N-1}\mathbf{1}_{N-1})^{-1}\mathbf{1}_{N-1}$.

Note that $\mathbf{1}_{N-1}\mathbf{1}_{N-1}^\mathsf{T}$ has one eigenvector $v_1 = \frac{1}{\sqrt{N-1}}\mathbf{1}$ with eigenvalue $\lambda_1 = N - 1$, and the remaining eigenvalues are all zero with eigenvectors satisfying $v_i^\mathsf{T}\mathbf{1} = 0$. Adding $I$ to this matrix simply increases all eigenvalues by one. Thus

$$\left(I + \mathbf{1}\mathbf{1}^\mathsf{T}\right)^{-1}\mathbf{1} = \frac{1}{N}\left(\frac{1}{\sqrt{N-1}}\mathbf{1}\right)\left(\frac{1}{\sqrt{N-1}}\mathbf{1}\right)^\mathsf{T}\mathbf{1} + \sum_{i=2}^{N-1} v_i \underbrace{v_i^\mathsf{T}\mathbf{1}}_{0} = \frac{1}{N}\underbrace{\frac{\mathbf{1}^\mathsf{T}\mathbf{1}}{N-1}}_{1}\mathbf{1} = \frac{1}{N}\mathbf{1},$$

and so $\mu = \frac{1}{N}\mathbf{1}_{N(N-1)}$, which is indeed $\geq \mathbf{0}$; thus this is an optimal solution to the problem.

We next reconstruct $\mathbf{w} = -A^\mathsf{T}\mu = -\frac{1}{N}A^\mathsf{T}\mathbf{1}_{N(N-1)}$. Consider the block $w_i$ inside $\mathbf{w}$; its value will be the negative mean of the entries of $A$ with an $E_i$ in them. The $\ell(i, j)$ rows for $j \neq i$ contribute $N - 1$ entries of the form $-E_i x_i$. We also have the $\ell(k, i)$ rows, which have one $E_i x_k$ term for each $k \neq i$. Thus

$$w_i = -\frac{1}{N}\left(-(N-1)x_i + \sum_{k \neq i} x_k\right) = -\frac{1}{N}\left(-Nx_i + \sum_{k=1}^{N} x_k\right) = x_i - \bar{x},$$

where $\bar{x} = \frac{1}{N}\sum_{k=1}^{N} x_k$. Thus, for a test point $x$,

$$\arg\max_i w_i^\mathsf{T} x = \arg\max_i x_i^\mathsf{T} x - \bar{x}^\mathsf{T} x = \arg\max_i x_i^\mathsf{T} x.$$

Because the $x_i$ are orthonormal, this is further equal to

$$\arg\min_i \|x_i\|^2 + \|x\|^2 - 2x_i^\mathsf{T} x = \arg\min_i \|x - x_i\|. \qquad \square$$

**Lemma 2.** *If $X \mid Y = y \sim \mathcal{N}(\mu_y, \sigma^2 I)$ and $Y \sim \mathrm{Uniform}([N])$, the Bayes-optimal classifier is given by*

$$f^*(x) = \arg\min_y \|x - \mu_y\|.$$

*Proof.* This well-known fact follows by combining

$$p(Y = y \mid X = x) = \frac{p(X = x \mid Y = y)p(Y = y)}{p(X = x)} \propto p(X = x \mid Y = y)$$

with the definition of the density for $X$,

$$\arg\max_y \frac{1}{(2\pi\sigma^2)^{d/2}}\exp\left(-\frac{1}{2\sigma^2}\|x - \mu_y\|^2\right) = \arg\min_y \|x - \mu_y\|. \qquad \square$$

## B  IMPLEMENTATION DETAILS FOR META LEARNING ALGORITHMS

**Metric-based**  We use a meta learning library called learn2learn (Arnold et al., 2020) to implement **ProtoNet** (Snell et al., 2017). Following the original paper, we train a model with 30-way and 20-way for 1-Shot and 5-Shot, respectively, for 3,000 iterations. We use a 4 layer convolutional neural network (Conv4) with 64 channel size, and the batch size is set to 100. For optimization, we employ an Adam optimizer with a learning rate of 0.01 without having a learning rate schedule.

**Optimization-based**  We use the learn2learn library to implement both **MAML** (Finn et al., 2017) and **ANIL** (Raghu et al., 2020). We use Conv4 with 32 channel size for MAML and 64 channel size for ANIL (larger channel size does not perform better for MAML). We train both MAML and ANIL for 60,000 iterations. For optimizer, we employ an Adam optimizer for both with learning rates of 0.003 and 0.001 (adaptation learning rates of 0.5 and 0.1) for MAML and ANIL, respectively. Batch sizes are set to 32 for both.

For **MetaOptNet** (Lee et al., 2019b), we use the publicly available code provided by the authors of the paper (`https://github.com/kjunelee/MetaOptNet`). We employ the dual formulation of Support Vector Machine (SVM) proposed in MetaOptNet (MetaOptNet-SVM) for experiments with the training shot of 15, and use the default hyperparameter settings. For instance, we use a SGD optimizer with initial learning rate of 0.1 which decays step-wise. We train a model for 60 epochs with a batch size of 8.

**Model-based**  For both Conditional Neural Process (CNP) (Garnelo et al., 2018a) and Attentive Neural Process (ANP) (Kim et al., 2019), we use the publicly available code provided by the authors of the paper that addresses regression tasks for computer vision problems (Gao et al., 2022) (`https://github.com/boschresearch/what-matters-for-meta-learning`).

As the authors provide the model checkpoints for CNP on Distractor dataset and ANP on ShapeNet1D, we utilize them to compare active learning methods in meta-test time. We use 2-Shot for context sets in meta-test time instead of 25-Shot as done in the original work, since 25-Shot is too large to investigate the difference between active learning methods.

**Pre-training-based**  We use the publicly available code provided by the authors of the papers for both **Baseline++** (Chen et al., 2019) (`https://github.com/wyharveychen/CloserLookFewShot`) and **SimpleShot** (Wang et al., 2019) (`https://github.com/mileyan/simple_shot`). For both models, we use the features from the pre-trained models on the whole training dataset in inference time. As reported in the public repository for Baseline++, the performance on CUB for 1-Shot and 5-Shot is lower than the numbers reported in the paper by about 1.1% and 2.5%, respectively. Similarly, the reproduced performance of SimpleShot for 1-Shot and 5-Shot is lower by about $4 \sim 5\%$. Note that the numbers correspond for the case of fully stratified random sampling.

## C  IMPLEMENTATION DETAILS FOR ACTIVE LEARNING STRATEGIES

In this section, we provide detailed description for the implementation of the following active learning methods.

**DPP (Bıyık et al., 2019)** We use DPPy library (Gautier et al., 2019) to implement DPP selection. Gram matrix of the features from the penultimate layer are used as L-ensembles for DPP. We employ $k$-DPP to select $k$ number of context data points.

**Coreset (Sener & Savarese, 2018)** We refer to both original code (`https://github.com/ozansener/active_learning_coreset`) and code provided by the authors of Typiclust and ProbCover. Since we assume that there is no initial labeled data points, we randomly choose the first data point and then apply the greedy algorithm after that.

**Typiclust (Hacohen et al., 2022)** We refer to the publicly available code provided by the authors of the paper (`https://github.com/avihu111/TypiClust`). As the maximum number of data points to annotate is 25 (= 5-Way $\times$ 5-Shot), we do not set the maximum number of clusters unlike the original paper. We set the $k$ in $k$-NN to 20 as with the original work.

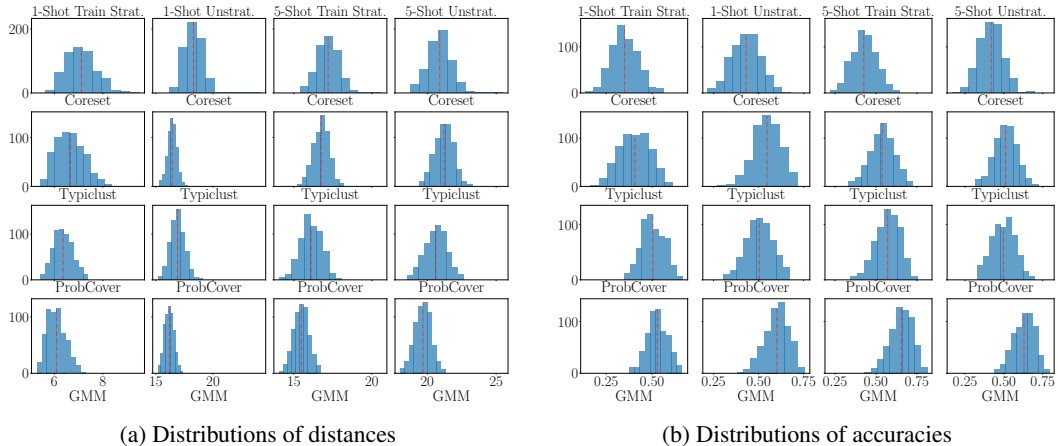

(a) Distributions of distances        (b) Distributions of accuracies

Figure 6: Estimation of goodness of selected data points on MiniImageNet with ANIL using the distribution of (a) the distance between the unlabeled points and closest selected points, and (b) the equality between the true labels of unlabeled points and labels of the closest select points. Red dotted lines show mean values.

| Data & Model | Clustering | 1-Shot | | | 5-Shot | | |
|---|---|---|---|---|---|---|---|
| | | Fully strat. | Train strat. | Unstrat. | Fully strat. | Train strat. | Unstrat. |
| MiniImage. MAML | $k$-means | $56.75 \pm 0.20$ | $\mathbf{33.29 \pm 0.26}$ | $37.26 \pm 0.18$ | $65.76 \pm 0.18$ | $41.61 \pm 0.24$ | $\mathbf{59.17 \pm 0.20}$ |
| | $k$-means++ | $56.12 \pm 0.26$ | $32.87 \pm 0.32$ | $\mathbf{38.53 \pm 0.21}$ | $65.49 \pm 0.21$ | $43.61 \pm 0.32$ | $58.63 \pm 0.26$ |
| | GMM | $\mathbf{58.82 \pm 0.24}$ | $33.34 \pm 0.24$ | $37.68 \pm 0.19$ | $\mathbf{67.18 \pm 0.18}$ | $54.35 \pm 0.20$ | $59.05 \pm 0.20$ |
| FC100 ProtoNet | $k$-means | $\mathbf{50.20 \pm 0.17}$ | $29.69 \pm 0.20$ | $\mathbf{35.03 \pm 0.23}$ | $54.07 \pm 0.17$ | $41.42 \pm 0.23$ | $41.34 \pm 0.23$ |
| | $k$-means++ | $49.91 \pm 0.17$ | $27.27 \pm 0.22$ | $34.93 \pm 0.27$ | $54.72 \pm 0.30$ | $41.61 \pm 0.39$ | $42.64 \pm 0.39$ |
| | GMM | $50.22 \pm 0.18$ | $\mathbf{34.23 \pm 0.23}$ | $\mathbf{35.03 \pm 0.23}$ | $\mathbf{54.76 \pm 0.17}$ | $\mathbf{46.30 \pm 0.21}$ | $\mathbf{47.03 \pm 0.20}$ |

Table 6: Comparison of GMM and $k$-Means selections on MiniImageNet and FC100 using MAML and ProtoNet.

**ProbCover (Yehuda et al., 2022)** We use the code provided by the original authors of the paper (it is the same as Typiclust). As we state in Appendix F and Appendix I, we exploit the features from the meta learners instead of self-supervised features to determine the radius parameters of ProbCover. In particular, the radius for each algorithm and dataset combination is determined as shown in Appendix F.

**GMM (Ours)** We refer to a publicly available implementation for GMM (`https://github.com/ldeecke/gmm-torch`). As previously mentioned, we initialize the cluster centers using $k$-means. Then, we update the cluster means and covariance matrix (shared by all the clusters) using expectation maximization algorithm for up to 100 iterations. We make the covariance matrix shared between the clusters because we assume that the "influence" of each annotated data point to other data points is roughly the same regardless of data point although the weight of each dimension may be different (if they are the same, it is equivalent to $k$-means).

## D COMPARISON OF QUALITY OF SELECTED DATA POINTS

In this section, we estimate the quality of selected data points from the low budget active learning methods. In Figure 6, we compare them in the distance and accuracy as explained in the caption with ANIL (Raghu et al., 2020) on MiniImageNet. Whether a task is 1-Shot or 5-Shot, or train-time stratified or unstratified, we can observe that the metrics for GMM are consistently the best.

## E COMPARISON TO $k$-MEANS BASED METHODS

Table 6 compares the proposed GMM method to $k$-means and weighted $k$-means (Nguyen & Smeulders, 2004; Donmez et al., 2007) on MiniImageNet and FC100 datasets with MAML and ProtoNet, respectively.

Nguyen & Smeulders (2004); Donmez et al. (2007) combines uncertainty and density-based sampling *e.g.* $k$-means for binary classification using logistic regression. For their original tasks, the selection criterion is equivalent to density weighted margin sampling. We extend it to $K$ way multi-class classification tasks where their acquisition criterion becomes *weighted entropy*.

For most cases, weighted entropy is constantly worse than the other two methods. It is expected since uncertainty-based criteria perform significantly worse than density-based criteria throughout all the experiments.

Also, the performance of GMM and $k$-means is similar in general but for some cases, GMM is significantly better than $k$-means. We conjecture it is because some features are more important than the others, and since GMM takes it into account using Mahalanobis distance (instead of Euclidean distance used in $k$-means), it selects data points that represents nearby data points better.

## F    DIFFICULTY OF TUNING THE RADIUS PARAMETER FOR PROBCOVER

In Section 3.2 of Yehuda et al. (2022), the authors proposed to tune the radius $\delta$ based on the purity defined as,

$$\pi(\delta) = P(\{x : B_\delta(x) \text{ is pure}\}) \quad \text{where} \quad B_\delta(x) = \{x' : \|x' - x\|_2 \leq \delta\} \tag{5}$$

Here, a ball $B_\delta(x)$ is "pure" if $f(x') = y, \forall x' \in B_\delta(x)$ where $y$ is the label of $x$. As the radius $\delta$ increases, the purity decreases monotonically. They choose the optimal radius $\delta^*$ as $\delta^* = \max\{\delta : \pi(\delta) \geq 0.95\}$. More specifically, they first run k-means with k being the number of classes. Then, the purity is measured using the k-means assignment as pseudo-labels.

In their setting (pool-based active learning for image classification), since it is hard to obtain meaningful features from a model trained only a few examples, they use the features from self-supervised learning methods such as SimCLR (Chen et al., 2020) It is, however, not the case for meta-learning. In meta-test time, the features from the meta learner are usually more meaningful than self-supervised learning features. Hence, we use the mete learner's features to estimate the optimal radius for ProbCover. Following the original paper, we first run k-means and compute the purity in the same way. Since the features can differ by meta learning algorithms and the number of shots, we provide the plots for different algorithms as well as 1 and 5-Shots as shown in Figure 7 (we select the optimal radius $\delta$ based on these plots throughout the experiments). For Figure 7(a)-(f), we also provide the meta-test performance of stratified and unstratified versions of Random selection to demonstrate that the estimated optimal radius and best radius for meta-test accuracy do not align.

Another difficulty of estimating the optimal radius is that it is hard to set a search space for the radius. As shown in the x-axis of Figure 7, the reasonable search space varies significantly depending on the meta-learning algorithms and datasets we use. In Yehuda et al. (2022), this was less of a problem since they use SimCLR features, which are normalized: the range of the radius is in $[0, 1]$. However, as shown in Appendix I, if we use SimCLR features in meta-test time to actively select context sets, the performance generally drops.

## G    ADDITIONAL EXPERIMENTAL RESULTS FOR CLASSIFICATION

In this section, we provide addtional experimental results for few-shot image classification. In Table 7, we compare the active learning strategies for ANIL (Raghu et al., 2020) on the TieredImageNet dataset. Similarly, Table 8 provides the results with MetaOptNet (Lee et al., 2019b) on FC100 dataset. Table 9, Table 10, and Table 11 are for SimpleShot (Wang et al., 2019), ProtoNet (Snell et al., 2017), and ANIL (Raghu et al., 2020) on MiniImageNet, respectively. Note that Entropy and Margin selections are not applicable for MetaOptNet-SVM. Regardless of meta-learning algorithm and dataset, GMM significantly outperforms the other active learning methods, and some of them are worse than the Random selection.

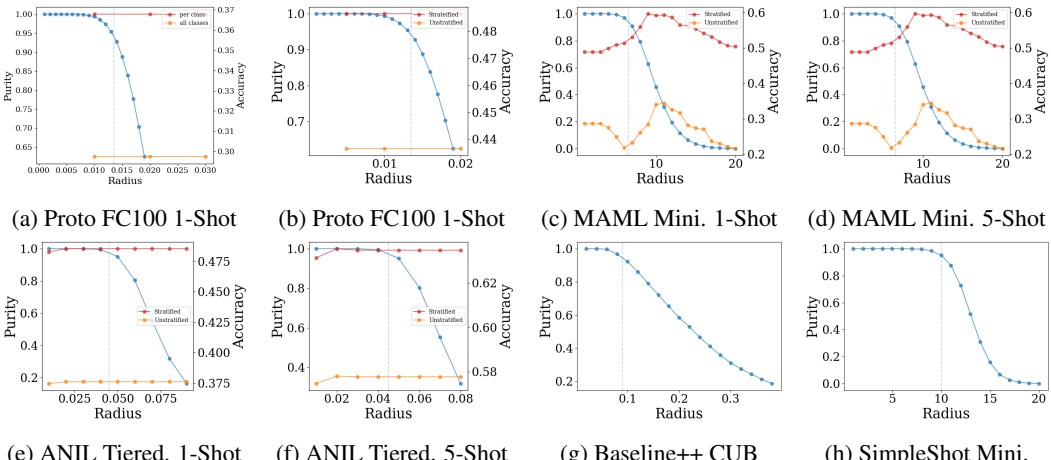

(a) Proto FC100 1-Shot     (b) Proto FC100 1-Shot     (c) MAML Mini. 1-Shot     (d) MAML Mini. 5-Shot

(e) ANIL Tiered. 1-Shot     (f) ANIL Tiered. 5-Shot     (g) Baseline++ CUB     (h) SimpleShot Mini.

Figure 7: Estimation of the optimal radius for ProbCover in meta-learning

| $\text{Pick}_\theta^{eval}$ | 1-Shot | | | 5-Shot | | |
|---|---|---|---|---|---|---|
| | Fully strat. | Train strat. | Unstrat. | Fully strat. | Train strat. | Unstrat. |
| Random | $47.55 \pm 0.18$ | $\mathbf{38.19 \pm 0.16}$ | $34.79 \pm 0.15$ | $\mathbf{63.84 \pm 0.17}$ | $\mathbf{57.92 \pm 0.23}$ | $\mathbf{57.56 \pm 0.18}$ |
| Entropy | $43.89 \pm 0.16$ | $32.73 \pm 0.16$ | $26.33 \pm 0.14$ | $57.56 \pm 0.18$ | $40.23 \pm 0.18$ | $34.16 \pm 0.17$ |
| Margin | $47.35 \pm 0.17$ | $36.01 \pm 0.14$ | $30.79 \pm 0.14$ | $62.87 \pm 0.17$ | $54.89 \pm 0.24$ | $56.76 \pm 0.17$ |
| DPP | $\mathbf{49.28 \pm 0.17}$ | $\mathbf{38.17 \pm 0.15}$ | $\mathbf{36.52 \pm 0.15}$ | $63.24 \pm 0.19$ | $57.28 \pm 0.21$ | $\mathbf{57.23 \pm 0.18}$ |
| Coreset | $47.32 \pm 0.18$ | $36.97 \pm 0.20$ | $\mathbf{40.72 \pm 0.14}$ | $56.93 \pm 0.18$ | $47.68 \pm 0.22$ | $52.89 \pm 0.17$ |
| Typiclust | $\mathbf{52.95 \pm 0.18}$ | $37.21 \pm 0.17$ | $34.05 \pm 0.14$ | $63.13 \pm 0.19$ | $55.84 \pm 0.22$ | $56.76 \pm 0.17$ |
| ProbCover | $48.53 \pm 0.53$ | $37.61 \pm 0.49$ | $34.53 \pm 0.43$ | $\mathbf{63.48 \pm 0.51}$ | $\mathbf{57.77 \pm 0.56}$ | $57.12 \pm 0.58$ |
| GMM (Ours) | $\mathbf{60.29 \pm 0.19}$ | $\mathbf{50.92 \pm 0.22}$ | $\mathbf{42.17 \pm 0.17}$ | $\mathbf{66.48 \pm 0.18}$ | $\mathbf{60.12 \pm 0.24}$ | $\mathbf{60.28 \pm 0.17}$ |

Table 7: 5-Way K-Shot classification on TieredImageNet with ANIL, with $\text{Pick}_\theta^{train}$ random.

## H  ADDITIONAL EXPERIMENTAL DETAILS FOR REGRESSION

Gao et al. (2022) propose the Distractor and ShapeNet1D datasets to compare meta learning algorithms for vision regression tasks. They evaluate meta learners for intra-category (IC) and cross-category (CC) inputs where CC corresponds to the cross-domain in few-shot image classification.

**Distractor** consists of 10 object classes for a training set and 2 novel classes for CC evaluation. Each class contains $1,000$ randomly sampled objects from ShapeNetCoreV2 (Chang et al., 2015). $20\%$ of training set is reserved for IC evaluation. In this dataset, each image consists of two objects: the object of interest and a distractor object, which are positioned randomly. The goal is to recognize and locate the object of interest within the image in the presence of a distractor.

**ShapeNet1D** (Gao et al., 2022) consists of 27 object classes for a training set and 3 object classes for CC evaluation. Each object class contains 50 images, and 10 images are used for IC evaluation. ShapeNet1D aims to predict the 1D pose, i.e., rotation angle, around the azimuth axis of an object.

To analyze these vision regression tasks, we compare various active learning strategies in the 2-shot setting. We use CNP for Distractor, NP for ShapeNet1D. More details about the models can be found in Appendix B.

## I  COMPARISON TO SELF-SUPERVISED FEATURES

ProbCover and Typiclust use self-supervised features to actively select new data points to annotate, since there are not enough labeled data to train a classifier to output meaningful features. Instead, they utilize the features from SimCLR (Chen et al., 2020). To validate if it is better to use the features from a meta learner than SimCLR in meta-learning, we compare SimCLR features to the features from either MAML or ProtoNet for Typiclust and ProbCover as shown in Table 12 and Table 13.

| $\text{Pick}_\theta^{eval}$ | 1-Shot | | | 5-Shot | | |
|---|---|---|---|---|---|---|
| | Fully strat. | Train strat. | Unstrat. | Fully strat. | Train strat. | Unstrat. |
| Random | $40.41 \pm 0.74$ | $31.96 \pm 0.56$ | $32.76 \pm 0.63$ | $53.11 \pm 0.66$ | $47.73 \pm 0.70$ | $47.48 \pm 0.76$ |
| DPP | $40.47 \pm 0.80$ | $30.33 \pm 0.67$ | $33.41 \pm 0.66$ | $51.44 \pm 0.68$ | $48.21 \pm 0.67$ | $47.45 \pm 0.68$ |
| Coreset | $39.20 \pm 0.71$ | $27.55 \pm 0.66$ | $30.16 \pm 0.69$ | $46.80 \pm 0.67$ | $24.08 \pm 0.65$ | $25.75 \pm 0.72$ |
| Typiclust | $45.20 \pm 0.78$ | $26.35 \pm 0.47$ | $27.00 \pm 0.43$ | $52.39 \pm 0.66$ | $23.97 \pm 0.42$ | $24.12 \pm 0.39$ |
| ProbCover | $41.93 \pm 0.67$ | $26.87 \pm 0.62$ | $27.43 \pm 0.48$ | $54.36 \pm 0.76$ | $37.00 \pm 0.69$ | $38.33 \pm 0.76$ |
| GMM (Ours) | $51.16 \pm 0.67$ | $40.89 \pm 0.74$ | $41.61 \pm 0.87$ | $60.48 \pm 0.86$ | $52.68 \pm 0.70$ | $51.79 \pm 0.70$ |

Table 8: 5-Way K-Shot classification on FC100 with MetaOptNet, with $\text{Pick}_\theta^{train}$ random.

| $\text{Pick}_\theta^{eval}$ | 1-Shot | | 5-Shot | |
|---|---|---|---|---|
| | Fully strat. | Train strat. | Fully strat. | Train strat. |
| Random | $45.15 \pm 0.73$ | $26.28 \pm 0.61$ | $61.22 \pm 0.72$ | $51.89 \pm 0.73$ |
| Entropy | $37.08 \pm 0.75$ | $21.62 \pm 0.37$ | $47.93 \pm 0.74$ | $32.74 \pm 0.60$ |
| Margin | $41.53 \pm 0.73$ | $24.28 \pm 0.51$ | $62.15 \pm 0.70$ | $50.90 \pm 0.75$ |
| DPP | $44.52 \pm 0.75$ | $26.32 \pm 0.58$ | $60.93 \pm 0.72$ | $51.79 \pm 0.75$ |
| Coreset | $45.85 \pm 0.73$ | $27.04 \pm 0.54$ | $56.48 \pm 0.72$ | $40.39 \pm 0.68$ |
| Typiclust | $44.53 \pm 0.71$ | $22.97 \pm 0.42$ | $34.21 \pm 0.77$ | $20.04 \pm 0.06$ |
| ProbCover | $49.32 \pm 0.71$ | $24.61 \pm 0.52$ | $55.60 \pm 0.66$ | $32.24 \pm 0.67$ |
| GMM (Ours) | $52.77 \pm 0.72$ | $28.17 \pm 0.64$ | $62.64 \pm 0.71$ | $50.40 \pm 0.75$ |

Table 9: 5-Way K-Shot classification on MiniImageNet with SimpleShot, with $\text{Pick}_\theta^{train}$ random.

| $\text{Pick}_\theta^{eval}$ | 1-Shot | | | 5-Shot | | |
|---|---|---|---|---|---|---|
| | Fully strat. | Train strat. | Unstrat. | Fully strat. | Train strat. | Unstrat. |
| Random | $47.70 \pm 0.20$ | $39.65 \pm 0.28$ | $38.72 \pm 0.27$ | $64.66 \pm 0.18$ | $57.36 \pm 0.27$ | $57.42 \pm 0.25$ |
| Entropy | $44.33 \pm 0.20$ | $36.35 \pm 0.28$ | $34.87 \pm 0.27$ | $61.23 \pm 0.19$ | $49.83 \pm 0.31$ | $48.46 \pm 0.32$ |
| Margin | $47.07 \pm 0.20$ | $37.69 \pm 0.27$ | $37.84 \pm 0.28$ | $63.79 \pm 0.18$ | $55.25 \pm 0.29$ | $56.15 \pm 0.27$ |
| DPP | $47.90 \pm 0.20$ | $39.17 \pm 0.28$ | $37.89 \pm 0.26$ | $64.36 \pm 0.19$ | $57.48 \pm 0.26$ | $57.37 \pm 0.25$ |
| Coreset | $47.86 \pm 0.20$ | $39.51 \pm 0.26$ | $37.79 \pm 0.26$ | $55.09 \pm 0.20$ | $50.14 \pm 0.29$ | $50.27 \pm 0.28$ |
| Typiclust | $59.51 \pm 0.17$ | $38.47 \pm 0.27$ | $37.57 \pm 0.27$ | $61.02 \pm 0.19$ | $51.82 \pm 0.31$ | $52.02 \pm 0.30$ |
| ProbCover | $48.51 \pm 0.20$ | $35.25 \pm 0.26$ | $34.50 \pm 0.25$ | $43.61 \pm 0.19$ | $38.63 \pm 0.21$ | $38.24 \pm 0.20$ |
| GMM (Ours) | $64.50 \pm 0.16$ | $47.88 \pm 0.32$ | $44.71 \pm 0.29$ | $67.03 \pm 0.19$ | $57.55 \pm 0.29$ | $56.44 \pm 0.30$ |

Table 10: 5-Way K-Shot classification on MiniImageNet with ProtoNet, with $\text{Pick}_\theta^{train}$ random. The **first**, **second**, **third** best results for each setting are marked in this and all other results tables.

| $\text{Pick}_\theta^{eval}$ | 1-Shot | | | 5-Shot | | |
|---|---|---|---|---|---|---|
| | Fully strat. | Train strat. | Unstrat. | Fully strat. | Train strat. | Unstrat. |
| Random | $46.59 \pm 0.19$ | $36.70 \pm 0.19$ | $34.79 \pm 0.18$ | $61.35 \pm 0.19$ | $55.24 \pm 0.20$ | $56.65 \pm 0.19$ |
| Entropy | $44.63 \pm 0.20$ | $35.51 \pm 0.18$ | $27.35 \pm 0.14$ | $55.09 \pm 0.19$ | $39.71 \pm 0.20$ | $37.45 \pm 0.19$ |
| Margin | $46.58 \pm 0.19$ | $36.60 \pm 0.19$ | $32.46 \pm 0.18$ | $55.62 \pm 0.19$ | $40.40 \pm 0.20$ | $37.67 \pm 0.19$ |
| DPP | $47.33 \pm 0.19$ | $37.45 \pm 0.17$ | $37.76 \pm 0.18$ | $61.08 \pm 0.19$ | $56.18 \pm 0.18$ | $57.08 \pm 0.18$ |
| Coreset | $46.40 \pm 0.21$ | $38.37 \pm 0.17$ | $41.34 \pm 0.17$ | $53.74 \pm 0.20$ | $47.81 \pm 0.20$ | $51.62 \pm 0.19$ |
| Typiclust | $54.44 \pm 0.18$ | $36.78 \pm 0.17$ | $34.52 \pm 0.19$ | $60.87 \pm 0.18$ | $52.56 \pm 0.20$ | $55.11 \pm 0.19$ |
| ProbCover | $51.56 \pm 0.18$ | $27.49 \pm 0.15$ | $41.46 \pm 0.17$ | $61.68 \pm 0.18$ | $53.80 \pm 0.20$ | $42.70 \pm 0.22$ |
| GMM (Ours) | $58.50 \pm 0.18$ | $48.13 \pm 0.20$ | $40.26 \pm 0.18$ | $65.14 \pm 0.17$ | $59.01 \pm 0.20$ | $61.48 \pm 0.19$ |

Table 11: 5-Way K-Shot classification on MiniImageNet with ANIL, with $\text{Pick}_\theta^{train}$ random.

Here, we use MiniImageNet and FC100 datasets for MAML and ProtoNet, respecitvely as with Table 2 and Table 1. For both Typiclust and ProbCover, although there are a couple of cases where SimCLR features are better, it is significantly worse than MAML and ProtoNet features in general. It intuitively makes sense because 1) meta learners are trained with large enough data points and 2) it is likely that the information in self-supervised features do not align with that in meta learners.

| Dataset | Features | 1-Shot | | | 5-Shot | | |
|---|---|---|---|---|---|---|---|
| | | Fully strat. | Train strat. | Unstrat. | Fully strat. | Train strat. | Unstrat. |
| Mini. | MAML | **55.65 ± 0.18** | 27.45 ± 0.17 | **35.46 ± 0.18** | 64.16 ± 0.18 | **46.70 ± 0.21** | **57.83 ± 0.21** |
| | SimCLR | 44.84 ± 0.44 | **27.59 ± 0.35** | 34.80 ± 0.47 | **65.95 ± 0.43** | 36.03 ± 0.48 | 57.77 ± 0.47 |
| FC100 | ProtoNet | **46.01 ± 0.16** | **30.96 ± 0.19** | **30.61 ± 0.21** | 47.54 ± 0.17 | **43.61 ± 0.18** | **44.03 ± 0.21** |
| | SimCLR | 36.07 ± 0.44 | 29.60 ± 0.46 | 30.13 ± 0.45 | **48.59 ± 0.49** | 43.29 ± 0.49 | 43.89 ± 0.59 |

Table 12: Comparison of MAML and SimCLR features for Typiclust in meta-test time.

| Dataset | Features | 1-Shot | | | 5-Shot | | |
|---|---|---|---|---|---|---|---|
| | | Fully strat. | Train strat. | Unstrat. | Fully strat. | Train strat. | Unstrat. |
| Mini. | MAML | **52.81 ± 1.16** | 21.91 ± 0.24 | **36.21 ± 0.18** | **64.70 ± 0.91** | **42.07 ± 0.49** | 23.40 ± 0.36 |
| | SimCLR | 47.57 ± 0.42 | **25.35 ± 0.38** | 32.19 ± 0.43 | 64.33 ± 0.39 | 36.64 ± 0.58 | **26.16 ± 0.43** |
| FC100 | ProtoNet | **48.66 ± 0.16** | **32.86 ± 0.22** | **33.58 ± 0.19** | **51.11 ± 0.17** | **44.20 ± 0.24** | 44.40 ± 0.24 |
| | SimCLR | 31.40 ± 0.42 | 29.53 ± 0.42 | 28.39 ± 0.43 | 47.11 ± 0.39 | 39.33 ± 0.54 | **45.40 ± 0.52** |

Table 13: Comparison of MAML and SimCLR features for ProbCover in meta-test time.

## J  TRAINING-TIME ACTIVE LEARNING

As mentioned in Section 5, we observe that active learning methods do not significantly change the generalization performance of meta learners when applied in meta-train time, which aligns with Ni et al. (2021) and Setlur et al. (2020). To empirically demonstrate the statement, we apply several active learning methods without stratification in the meta-train time for ProtoNet on MiniImageNet. Again, we report the mean and 95% confidence interval from 600 meta-test tasks. Figure 8 shows among Random, DPP and GMM selections, one is not significantly better than another although the Entropy selection is significantly worse than them.

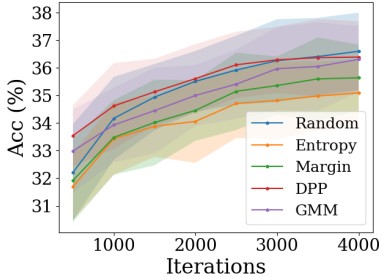

Figure 8: Comparison of $\text{Pick}_\theta^{train}$.

## K  SEQUENTIAL ACTIVE-META LEARNING

Although iterative sampling is more common in active learning, we have focused on sampling a context set at once because of the following two reasons,

- Even though we iterative label additional samples, the features do not change in most of meta-learning algorithms except for MAML. Even for other optimization-based methods such as ANIL, since the feature extractor is not updated during adaptation on a context set, the features will stay the same for iterative process of active learning. As we demonstrated with ProtoNet in Figure 9 (c)-(d) (details about experiments are below), although we iteratively add more labeled samples, the performance does not change much as the features do not change. In this case, selecting $N \times K$ samples at once is not different from iterative process while it is cheaper.

- If we iteratively add labeled samples, it will quickly go beyond few-shot regime in meta-learning, which is often not that practical in real world settings. Suppose we have a meta learner trained in 5-way 1-Shot. It is reasonable to add 5 samples per iteration since it is the minimum number to cover all the classes. But only after 5 iterations, it will go few-shot regime where we typically have 25 labeled context samples. It is even less practical for 5-Shot case.

Figure 9 compare active learning methods for sequential setting where we select 5 context samples at a time until the budget reaches 25 samples. Every time we select new context samples we may utilize them to maximize new label information. For MAML, we update all the model parameters through adaptation steps. It is, however, not applicable to the other meta-learning methods we use in

this work including ProtoNet, since none of the other methods including optimization-based methods such as ANIL, do not update the parameters up to the penultimate layer.

As expected, the test performance of ProtoNet does not change much regardless of active learning methods. But, the test performance of MAML gradually increases as we add more context samples. In sequential active-meta learning, GMM still significantly outperforms other active learning methods.

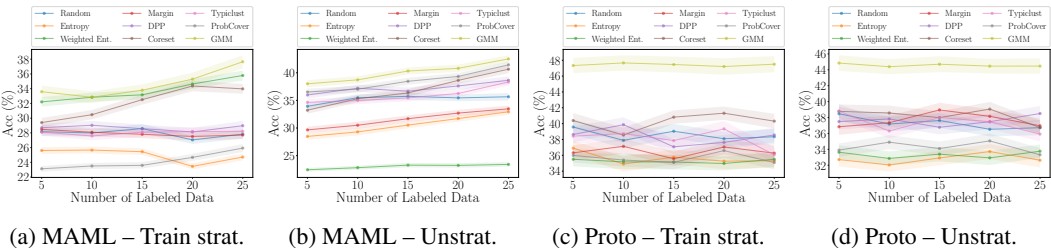

| (a) MAML – Train strat. | (b) MAML – Unstrat. | (c) Proto – Train strat. | (d) Proto – Unstrat. |

Figure 9: Test performance of MAML and ProtoNet on MiniImageNet with sequentially actively selected context sets. 5 context samples are selected at each iteration until it reaches 25.

## L    COMPARISON WITH HYBRID ACTIVE LEARNING METHODS

Table 14 shows that the proposed GMM-based method significantly outperforms all the hybrid methods - Weighted Entropy (Nguyen & Smeulders, 2004), BADGE (Ash et al., 2020), and Al-Shedivat et al. (2021).

- Nguyen & Smeulders (2004) proposed weighted expected error active learning method for binary classification but for multi-class classification, we derived that it becomes weighted entropy (we call Weighted Ent.) where weights are based on likelihood computed using soft $k$-means.

- BADGE (Ash et al., 2020) is one of popular hybrid active learning methods. It uses initialization of k-means++ where embedding is derived from the gradients of loss with respect to the output of the penultimate layer using pseudo labels.

- Al-Shedivat et al. (2021) first clusters unlabeled samples using $k$-means++ and selects samples per cluster using entropy.

This experiment along with the poor performance of uncertainty methods such as Entropy and Margin, demonstrates that for low budget regime, diversity measure is significantly more important than uncertainty measure.

**Discussion**    Most uncertainty measures should be high near decision boundaries, which is not desirable for low budget setting since those uncertain points tend to be outliers or are too hard to be generalized. In particular, the purpose of having context sets for meta-learning is to refer them when making predictions on target samples. For this, it is better to choose the points that are easy to refer. If selected context samples are too far away from the target samples, it would be hard to make good predictions for the target set. Diversity measure, especially GMM, ensures that the context set is not too far away from the target set even in the worst case (imagine some target samples are outliers). Therefore, it is desirable to only consider diversity for active selection of context sets in meta-learning. Hybrid methods considering both uncertainty and diversity may help for mid-budget active learning but it does not really help for extremely low budget scenario like meta-learning.

## M    FITTING GMM USING EXPECTATION MAXIMIZATION

In this section, we provide details about fitting GMM using the expectation maximization (EM) algorithm. Although it is available in many literature, we add it here for completeness of our method.

| Data & Model | Clustering | 1-Shot | | 5-Shot | |
|---|---|---|---|---|---|
| | | Train strat. | Unstrat. | Train strat. | Unstrat. |
| MiniImage. MAML | Weighted Ent. | $22.69 \pm 0.18$ | $32.27 \pm 0.32$ | $23.75 \pm 0.25$ | $46.80 \pm 0.33$ |
| | BADGE | $27.71 \pm 0.18$ | $34.30 \pm 0.21$ | $41.37 \pm 0.28$ | $58.79 \pm 0.24$ |
| | Al-Shedivat et al. | $30.59 \pm 0.28$ | $33.73 \pm 0.24$ | $38.24 \pm 0.29$ | $54.87 \pm 0.26$ |
| | GMM (Ours) | $\mathbf{33.34 \pm 0.24}$ | $\mathbf{37.68 \pm 0.19}$ | $\mathbf{54.35 \pm 0.20}$ | $\mathbf{59.05 \pm 0.20}$ |
| FC100 ProtoNet | Weighted Ent. | $31.80 \pm 0.20$ | $28.94 \pm 0.19$ | $40.40 \pm 0.25$ | $39.95 \pm 0.25$ |
| | BADGE | $30.91 \pm 0.23$ | $29.29 \pm 0.28$ | $43.85 \pm 0.22$ | $44.00 \pm 0.29$ |
| | Al-Shedivat et al. | $30.93 \pm 0.22$ | $30.43 \pm 0.24$ | $41.76 \pm 0.27$ | $43.41 \pm 0.29$ |
| | GMM (Ours) | $\mathbf{34.23 \pm 0.23}$ | $\mathbf{35.03 \pm 0.23}$ | $\mathbf{46.30 \pm 0.21}$ | $\mathbf{47.03 \pm 0.20}$ |

Table 14: Comparison of GMM with hybrid active learning methods.

The log-likelihood objective for a GMM is given by,

$$\ell(\theta) = \sum_{i=1}^{N} \log \left( \sum_{k=1}^{K} \pi_k \mathcal{N}(x_i|\mu_k, \Sigma_k) \right), \quad (6)$$

where model parameters $\theta = \{(\pi_k, \mu_k, \Sigma_k)\}_{k=1}^{K}$ with $N$ and $K$ being the number of samples and mixture components, respectively. EM algorithm is an iterative algorithm where we alternatively conduct E-step and M-step as follows,

- E-step: we compute the posterior probability $\omega_{ik}$ that represents $i$-th data point belongs to the $k$-th Gaussian component as,

$$w_{ik} = \frac{\pi_k \mathcal{N}(x_i|\mu_k, \Sigma_k)}{\sum_{j=1}^{K} \pi_j \mathcal{N}(x_i|\mu_j, \Sigma_j)} \quad (7)$$

- M-step: we maximize the log-likelihood in terms of the model parameters. Fortunately, for GMM, there are closed form solutions for each parameter.

$$\pi_k = \frac{1}{N} \sum_{i=1}^{N} w_{ik}, \quad \mu_k = \frac{\sum_{i=1}^{N} w_{ik} x_i}{\sum_{i=1}^{N} w_{ik}}, \quad \Sigma_k = \frac{\sum_{i=1}^{N} w_{ik}(x_i - \mu_k)(x_i - \mu_k)^T}{\sum_{i=1}^{N} w_{ik}} \quad (8)$$

We repeat the E-step and M-step until convergence of the log-likelihood or for a fixed number of iteration time. Please note that we use diagonal covariance $\Sigma_k$ since it is computationally efficient and often fits better in term of log-likelihood.

## N    More Analysis on Low Budget Active Learning Methods

We briefly explain why each active learning method does not perform as well as the proposed GMM method or even Random selection. In this section, we discuss further on the inferiority of low budget active learning methods compared to GMM. We conjecture it may be attributed to its implicit exploration of locally dense regions or inappropriate measure of representativeness. Here we analyzed potential reasons of their failure in very low budget regime.

- Typiclust: after conducting $k$-means, it selects samples for each cluster $j$, based on

$$\arg\max_{x \in \text{clust}_j} \left( \frac{1}{K} \sum_{x_i \in KNN(x)} ||x - x_i||_2 \right)^{-1}$$

where KNN denotes $k$-nearest neighbors of which size is fixed to 20. This measure seeks for locally dense region by selecting samples that are close to its nearest neighbors.

- ProbCover: it greedily finds the maximally covering samples given a fixed radius. This greedy algorithm provide $(1 - \frac{1}{e})$-approximation for the optimal solution but the gap with the optimal solution can be quite large. Also, the selection of the radius is hard as we discussed in Appendix F. When the radius is small, it tries to find samples that are in locally dense regions.

- DPP: it finds samples of which a kernel matrix (with a pre-defined kernel function) has the maximum determinant, which implicitly finds diverse samples. The determinant of a matrix, however, may not align with selecting maximum covering (or representative) samples. In particular, maximizing the determinant of the kernel matrix may lead to selecting samples far away from other samples.

Compared to these methods, GMM tries to find globally representative samples in non-greedy fashion (using expectation maximization). Also, its measure of covering other samples is in Mahalanobis distance, which intuitively makes more sense than the determinant of a kernel matrix as a measure. The proposed GMM method is also theoretically motivated by the Proposition 1, which says a classifier trained with the selected samples from GMM (cluster means) is a Bayes-optimal classifier under certain conditions.

