# OpenReview forum: "Exploring Active Learning in Meta-Learning: Enhancing Context Set Labeling"
_ICLR.cc/2024/Conference — Submitted to ICLR 2024_

### Official Review · Reviewer_r9Lv · 2023-10-29

**Soundness:** 3 good
**Presentation:** 3 good
**Contribution:** 2 fair
**Rating:** 6
**Confidence:** 3

**Summary:**

The paper focuses on active meta-learning. Firstly, it summarizes different ways of combining active learning with meta learning. Additionally, it suggested a new active learning methods based on Gaussian Mixture Selection for the testing stage of meta learning. The paper presents numerous experiments conducted on real-world datasets, comparing the results with baseline algorithms.

**Strengths:**

The paper gives a comprehensive summarization of the active selecting context in Meta learning, presentation and notation in this section is clear.

The paper provides numerous experiment results on both classification and regression problems.

**Weaknesses:**

The contribution of this paper is incremental. As the paper points out, Gaussian Mixture selection for active learning is not new. Although the paper combines it with meta learning, but the nothing is tailored for meta learning. To me, it simply applies existing algorithms in the testing stage of meta learning.

Some presentation in this paper is not good. For example, in the introduction, the paper does not explicitly highlight that the proposed method is designed for low budget, especially one-shot learning under an unstratified setting. In section 3.2, the paper proposes to use penultimate layer of the initialization neural net as the features for active context selection, but this important detail is only mentioned midway through that section.

The choice of Gaussian Mixture Selection is not adequately explained, and the paper fails to discuss its advantages compared to other clustering algorithms.

**Questions:**

In Figure 2, the caption mentions that the figure depicts mean and standard error, but in the figure itself, it actually represents accuracy.

In proposition 1, it should it be "X|Y =  x~N(mu_y, sigma^2 I)" ? And why this proposition is beneficial for active learning is not fully discussed.

In Figure 3, why is Gaussian Mixture Selection effective in covering more classes in the one-shot unstratified task? This appears to be a challenging task for unsupervised learning without the aid of labels. How does Gaussian Mixture Selection outperform other competing algorithms in this context?

---

> ### Author Response · Authors · 2023-11-20
> **Response to Reviewer r9Lv (1/3)**
>
> Thank you for your comments. Please see the common response above, in addition to discussion here.
>
>
> > The contribution of this paper is incremental. As the paper points out, Gaussian Mixture selection for active learning is not new. Although the paper combines it with meta learning, but the nothing is tailored for meta learning. To me, it simply applies existing algorithms in the testing stage of meta learning.
>
> We would first like to argue that the proposed GMM method is indeed particularly suited to meta-learning as we mentioned in the first response of the common response above. The reviewer will find more detailed answers there but here, we summarize it as follows:
>
>
> 1. We proposed the GMM method for very low budget regime where meta-learning belongs. The budgets of low budget active learning methods (around 60, 600, or 6000) are still much larger than that of typical meta learning where it is usually $\leq 25$.
> 2. The GMM method ensures that the context set is “good reference” for the target set, even in the worse case, as it selects the samples closest to the cluster means as the context set. It is opposite with uncertainty-based methods where they select samples in less dense regions. Even the low budget active learning methods select points in locally dense regions, not global coverage.
>
> We hope the reviewer recognizes that our work is specifically tailored for meta-learning. However, if there are still disagreements on this aspect, we would be more than willing to engage in further discussion to provide additional clarification or address any concerns..
>
>
>
>
> > Some presentation in this paper is not good. For example, in the introduction, the paper does not explicitly highlight that the proposed method is designed for low budget, especially one-shot learning under an unstratified setting.
>
> In the third paragraph of introduction, we specified that our focus is on active meta-learning and raised the question, "How can a meta-learner exploit an active learning setup to learn the best model possible, using only a very small number of labels in its context sets?”. In addition to this explicit reference to “a very small number of labels,” the vast majority of the extensive literature on meta-learning is for low-budget settings (one- or five-shot cases).
>
> We are willing to make this clearer if the reviewer still thinks it is not clearly highlighted enough, or to address any other points of presentation which are “not good.”
>
>
>
> > In section 3.2, the paper proposes to use penultimate layer of the initialization neural net as the features for active context selection, but this important detail is only mentioned midway through that section.
>
> First, we did briefly mention this point in the introduction: “In particular, we propose a natural algorithm based on fitting a Gaussian mixture model to the unlabeled data, using meta-learned feature representations.”
>
> Also, we frame active meta-learning in Section 2 and propose our method in Section 3. In Section 3.1, we review how previous works in active learning works, particularly focusing on low budget active learning since they are competing algorithms to ours. Then, in Section 3.2, we explained how features are selected in our method. Although feature selection is an important aspect of the algorithm, we think it makes sense to introduce how related methods work first. If you have a suggestion for a logical earlier place to discuss this place, however, we’re happy to take any suggestions.

---

> ### Author Response · Authors · 2023-11-20
> **Response to Reviewer r9Lv (2/3)**
>
> > The choice of Gaussian Mixture Selection is not adequately explained, and the paper fails to discuss its advantages compared to other clustering algorithms.
>
> One advantage of a very simple method is that it can be fully explained very briefly, which we do at the beginning of Section 3.3: “fit a mixture of $k$ Gaussians to the unlabeled data features, where $k$ is the label budget, using EM with a $k$-means initialization. We use a shared diagonal covariance matrix. Once a mixture is fit, we select the highest-density point from each component:
> $\arg \min_{x \in \mathcal U } (x - \mu_j)^T \Sigma^{-1} (x - \mu_j)$  for each $j \in [k]$.”
>
> For the sake of full completeness, we added more details about fitting Gaussian mixtures with EM to Appendix M.
>
> In terms of choosing Gaussian mixtures over other clustering algorithms, we indeed provided multiple empirical comparisons. Typiclust, based on $k$-means, is compared with in Tables 1-5 and 7-11. Table 6, in Appendix E, further directly compares Gaussian mixture modeling to $k$-means with exactly the same algorithm in two settings (MinImageNet with MAML and FC100 with ProtoNet). We further added a comparison to $k$-means++ initialization below, and updated Table 6 below. GMM generally either matches or significantly outperforms $k$-means methods, other than a single case where $k$-means++ performed slightly better (1-shot unstratified on MiniImageNet).
>
> Lastly, we also provided comparison with hybrid active learning methods, many of which have clustering components, in the common response as well as Appendix L.
>
> However, if the reviewer has particular clustering-based active learning methods that the reviewer thinks we need to compare ours with, please let us know. We are eager to conduct more experiments to enhance our work.
>
> | $Pick^{eval}_\theta$ | 1-Shot fully strat. | 1-Shot train strat. | 1-Shot unstrat.  | 5-Shot fully strat. | 5-Shot train strat. | 5-Shot unstrat.  |
> | -------------------- | ------------------- | ------------------- | ---------------- | ------------------- | ------------------- | ---------------- |
> | $k$-means            | 56.75 $\pm$ 0.20    | 33.29 $\pm$ 0.26    | 37.26 $\pm$ 0.18 | 65.76 $\pm$ 0.18    | 41.61 $\pm$ 0.24    | 59.17 $\pm$ 0.20 |
> | $k$-means++          | 56.12 $\pm$ 0.26    | 32.87 $\pm$ 0.32    | 38.53 $\pm$ 0.21 | 65.49 $\pm$ 0.21    | 43.61 $\pm$ 0.32    | 58.63 $\pm$ 0.26 |
> | GMM (ours)           | 58.82 $\pm$ 0.24    | 33.34 $\pm$ 0.24    | 37.68 $\pm$ 0.19 | 67.18 $\pm$ 0.18    | 54.35 $\pm$ 0.20    | 59.05 $\pm$ 0.20 |
>
> **Table. Comparison of the variants of $k$-means with GMM on MiniImageNet using MAML**
>
>
>
> | $Pick^{eval}_\theta$ | 1-Shot fully strat. | 1-Shot train strat. | 1-Shot unstrat.  | 5-Shot fully strat. | 5-Shot train strat. | 5-Shot unstrat.  |
> | -------------------- | ------------------- | ------------------- | ---------------- | ------------------- | ------------------- | ---------------- |
> | $k$-means            | 50.20 $\pm$ 0.17    | 29.69 $\pm$ 0.20    | 35.03 $\pm$ 0.23 | 54.07 $\pm$ 0.17    | 41.42 $\pm$ 0.23    | 41.34 $\pm$ 0.23 |
> | $k$-means++          | 49.91 $\pm$ 0.17    | 27.27 $\pm$ 0.22    | 34.93 $\pm$ 0.27 | 54.72 $\pm$ 0.30    | 41.61 $\pm$ 0.39    | 42.64 $\pm$ 0.39 |
> | GMM (ours)           | 50.22 $\pm$ 0.18    | 34.23 $\pm$ 0.23    | 35.03 $\pm$ 0.23 | 54.76 $\pm$ 0.17    | 46.30 $\pm$ 0.21    | 47.03 $\pm$ 0.20 |
>
> **Table. Comparison of the variants of $k$-means with GMM on FC100 using ProtoNet.**
>
>
>
> > In Figure 2, the caption mentions that the figure depicts mean and standard error, but in the figure itself, it actually represents accuracy.
>
> The values refer to the mean and standard error of the accuracy; we believe this was clear from context in the first place (the only other interpretation we can see being that they would refer to the mean features, which would not make sense), but we have made this more explicit in the revised paper.

---

> ### Author Response · Authors · 2023-11-20
> **Response to Reviewer r9Lv (3/3)**
>
> > In proposition 1, it should it be “$X|Y = x~N(\mu_y, \sigma^2 I)$" ? And why this proposition is beneficial for active learning is not fully discussed.
>
> We meant what we wrote: the random variable $X$, conditioned on the random variable $Y$ (the label) taking a specific value $y$ (e.g. 2), follows a Gaussian distribution with mean $\mu_y$ and covariance $\sigma^2 I$. We added parentheses to hopefully make the equation more unambiguous.
>
> In Section 3.3, right before the Proposition 1, we specified that “if class-conditional data distributions are isotropic Gaussians with the same covariance matrices, labeling the cluster centers can be far preferable to labeling a random point from each cluster.”
> This statement is backed up by Proposition 1: when we use a max-margin separator (as in MetaOptNet or, asymptotically, ANIL), training on cluster centers yields the Bayes-optimal classifier. This would not be the case with randomly selected data points. This is further illustrated by Figure 4 in Appendix A.
>
> For more general settings, we argue that GMM is still a good method based on being an efficient set cover (e.g. Figure 5 in Appendix A).
>
> If any aspects remain unclear, we welcome your further inquiries. Your feedback is invaluable, and we appreciate your thorough review.
>
>
>
> > In Figure 3, why is Gaussian Mixture Selection effective in covering more classes in the one-shot unstratified task? This appears to be a challenging task for unsupervised learning without the aid of labels. How does Gaussian Mixture Selection outperform other competing algorithms in this context?
>
> Thank you for bringing up an important question. We posit that the inferiority of the other competing methods compared to GMM may be attributed to its implicit exploration of locally dense regions or inappropriate measure of representativeness. Here we analyzed potential reasons of their failure in very low budget regime.
>
>
> - Typiclust: after conducting $k$-means, it selects samples for each cluster $j$, based on $ \arg \max_{x \in \mathcal clust_j} ( \frac{1}{K} \sum_{x_i \in KNN(x)} || x - x_i ||_2 ) ^{-1} $ where KNN denotes $k$-nearest neighbors of which size is fixed to 20. This measure seeks for locally dense region by selecting samples that are close to its nearest neighbors.
> - ProbCover: it greedily finds the maximally covering samples given a fixed radius. This greedy algorithm provide $(1 - \frac{1}{e})$-approximation for the optimal solution but the gap with the optimal solution can be quite large. Also, the selection of the radius is hard as we discussed in Appendix F. When the radius is small, it tries to find samples that are in locally dense regions.
> - DPP: it finds samples of which a kernel matrix (with a pre-defined kernel function) has the maximum determinant, which implicitly finds diverse samples. The determinant of a matrix, however, may not align with selecting maximum covering (or representative) samples. In particular, maximizing the determinant of the kernel matrix may lead to selecting samples far away from other samples.
>
> Compared to these methods, GMM tries to find globally representative samples in non-greedy fashion (using expectation maximization). Also, its measure of covering other samples is in Mahalanobis distance, which intuitively makes more sense than the determinant of a kernel matrix as a measure. The proposed GMM method is also theoretically motivated by the Proposition 1, which says a classifier trained with the selected samples from GMM (cluster means) is a Bayes-optimal classifier under certain conditions.

---

> > ### Comment · Reviewer_r9Lv · 2023-11-21
> >
> > I appreciate your thoughtful response to my feedback. Your arguments solve some of my concerns about the paper, especially they answer my last question, I'll raise my score to 6.

---

> > > ### Author Response · Authors · 2023-11-21
> > > **Response to Reviewer r9Lv**
> > >
> > > It is great to hear that the reviewer found our responses helpful, particularly for the last question. As it was helpful for the reviewer, we have added the response for the last question in Appendix N, as it would potentially help other readers to understand the phenomenon better. We appreciate the reviewer for the valuable input. If there are any further questions or concerns, we are eager to discuss them further. Thank you.

---

### Official Review · Reviewer_EFev · 2023-10-29

**Soundness:** 2 fair
**Presentation:** 3 good
**Contribution:** 1 poor
**Rating:** 3
**Confidence:** 3

**Summary:**

the abstract discusses the concept of active meta-learning, which involves actively selecting which data points to label in the context set during the meta-learning process. A proposed algorithm based on fitting Gaussian mixtures is introduced. The key findings suggest that this algorithm outperforms state-of-the-art active learning methods when used with various meta-learning algorithms across multiple benchmark datasets. In essence, the study highlights the potential advantages of integrating active learning principles into meta-learning to improve performance

**Strengths:**

Overall, the motivation of this paper is very interetsting to use active learning to save the label cost in meta learning. This paper summarize the difference of current related work and clarify their difference. I appreciated the conducted experiments.

**Weaknesses:**

The technique novelty is too limited. The introduced algrithm is very simple based on normal distribution. So, the overal paper looks more like a technique report or a survey.

**Questions:**

If the algorithm is theoretically motivated, why is there no theories or lemmas in the mainbody?

____
After rebuttal: I appreciate the author's response and added experiments. I would suggest adding more theoretical analysis to the main body for a simple method and moving a part of related work to the Appendix. Some short-paper tracks may be more suitable for this paper.

---

> ### Author Response · Authors · 2023-11-20
> **Response to Reviewer EFev**
>
> Thank you for your comments. Please see the common response above, in addition to discussion here.
>
>
> > The technique novelty is too limited. The introduced algrithm is very simple based on normal distribution. So, the overal paper looks more like a technique report or a survey.
>
>
> We agree that the proposed method is very simple, but we believe that algorithmic simplicity is not a limitation but rather a strength. As we have demonstrated throughout many experiments, our simple algorithm outperforms much more complicated existing active learning methods.
>
> More importantly, we would like to emphasize that our work is not merely an application of an existing method to a new task. The introduction of the GMM method stems from an observation that has not been considered in previous active learning research. As illustrated in Figure 2, the GMM method consistently matches or outperforms existing low-budget active learning methods in the very low budget regime where meta-learning is situated.
> Moreover, in the context of meta-learning, providing effective “reference points” is crucial. We propose the GMM method as it is well-suited for this purpose by ensuring that the context set remains proximate to the target set, even in worst-case scenarios.
>
> We have further listed contributions of our paper in the second response of the common response above. Here, we summarize our contributions as follows,
>
> 1. We provide a general framework for active meta-learning in meta-test time and identify a major challenge in the traditional setup for meta-learning in classification in Section 2.
> 2. We propose a simple active learning method tailored for meta-learning with an observation that is not considered in the previous works, along with theoretical motivation in Section 3.
> 3. We demonstrate the effectiveness and robustness of our method on multiple datasets with various meta-learning algorithms for both classification and regression tasks in Section 4.
>
>
> We sincerely hope the reviewer considers these contributions and re-evaluate our work. If the reviewer still has any remaining concerns or questions, we are eager to address them further.
>
>
> > If the algorithm is theoretically motivated, why is there no theories or lemmas in the mainbody?
>
> The theoretical motivation for our algorithm is prominently presented in the main body of the paper, specifically in Section 3.3 through Proposition 1. The proofs for Proposition 1 are rigorously established by Lemma 1 and Lemma 2, both of which can be found in Appendix A.1. While we recognize the importance of detailed theoretical foundations, due to space constraints, we had to place Lemma 1 and 2 in the appendix.

---

> ### Author Response · Authors · 2023-11-22
> **Response to the Post-Rebuttal Comment**
>
> We value your insightful feedback. After reviewing the post-rebuttal comment, we remain uncertain about the basis for the 'reject' rating.
>
> As highlighted in our common response and individual response to the reviewer, we explained that simplicity is a strength, not a weakness. However, if the reviewer still looks at it differently, we are open to further discussion.
>
> Additionally, we've included a theoretic motivation in Proposition 1, supported by a clear proof and additional insights in Appendix A.1. If the reviewer seeks more theoretical analysis, we kindly ask for specific aspect our work may lack in this regard.
>
> We kindly request a reconsideration of the overall rating. If the suggestion to move related work to the appendix is the remaining concern, we believe a 'reject' may be too severe. Your re-evaluation would be greatly appreciated.

---

### Official Review · Reviewer_PTqo · 2023-11-01

**Soundness:** 3 good
**Presentation:** 3 good
**Contribution:** 3 good
**Rating:** 6
**Confidence:** 4

**Summary:**

This paper studies the effectiveness of active learning in the meta-learning setting.  The authors first discuss the different ways active-learning can be applied to meta-learning including actively selecting data to label at training and test time as well as actively selecting tasks to train on.  Experiments with using active learning to select labeled data at meta-training time showed no benefit over uniform sampling so the authors focus on active labeling at meta-test time.  Here, they
propose a simple Gausian Mixture Model to identify highest-density points to label.  Experiments show GMM to outperform other labeling approaches at meta-test time on multiple computer vision meta-learning benchmarks.

**Strengths:**

- Active learning at meta-test time is a novel problem to study; prior work I am aware of have primarily focused on active learning during meta-training phase.
- Experimental results for GMM for active-learning at test time are quite strong compared to other approaches.
- Good coverage of meta-learning approaches including metric (ProtoNet), optimization (MAML), and model (Baseline++) type approaches.
- The paper is clear and easy to understand.

**Weaknesses:**

- Missing reference to [Al-Shedivant et al. 2021](https://arxiv.org/pdf/2102.00127.pdf).  This paper studies active learning for meta-learning at training time and proposes a hybrid informative and diverse clustering labeling approach using k-means++.  I encourage the authors to include this active learning approach as an additional baseline in their experiments.
- Limited technical novelty since theory and approach are straightforward.  However, I am not placing too much weight on this since experimental results are strong.
- The paper can benefit from a discussion of practical implications of being able to be more label-efficient at meta-test time grounded in a real-world example.
- Theoretical justification is very basic and presumed to hold in meta-learning case with ad-hoc justification.

**Questions:**

- For Tables 1, 2, 3, how many runs are used to compute error bars?
- Is there a hypothesis for why GMM helps at test time but not at meta-train time?  This is an important discrepancy that is worth understanding deeper.

---

> ### Author Response · Authors · 2023-11-20
> **Response to Reviewer PTqo (1/2)**
>
> Thank you for your comments. Please see the common response above, in addition to discussion here.
>
>
> > Missing reference to Al-Shedivant et al. 2021. […] I encourage the authors to include this active learning approach as an additional baseline in their experiments.
>
> Thanks for alerting us to this paper. Although they indeed focus on active learning in meta-training time, it is quite related to our work. We compared experimentally to their approach (along with other hybrid methods); results are discussed in the common response, but in short we substantially outperform their approach and other hybrid methods, probably because uncertainty methods perform so poorly in this very-low-budget regime.
>
> Most uncertainty measures should be high near decision boundaries and/or outlier points; this is undesirable for low-budget settings, since they are hard to generalize from in “initial” stages of learning. Methods like our GMM-based proposal ensure that the context set points are “near” most of the target set, helping in the early phases of learning which are all that meta-learning considers. Hybrid methods considering both uncertainty and diversity are likely to help in middle-budget regimes, but not in extremely-low budget settings like meta-learning. We’ve added discussion of this point to Appendix L of the revised paper.
>
> Note that for this experiment, we applied their method only in meta-test time, for fair comparison to other approaches. Applying this category of approach in meta-training, including theirs as well as ours, requires a substantial computational overhead, since we must run clustering for each task considered in meta-training; we saw in Appendix J that doing so does not seem helpful in general.
>
> We provide further analysis on Al-Shedivat et al. in the reviewer’s last question.
>
>
> > Limited technical novelty since theory and approach are straightforward. However, I am not placing too much weight on this since experimental results are strong.
>
> We appreciate the reviewer for valuing our strong experimental results. We agree that our proposed method is simple and straightforward, however, we want to emphasize that simple methods are not a limitation but rather a strength. Many existing active learning approaches, despite their improvement, require tuning several hyperparameters and complicated implementation details. For instance, as we discussed in Appendix F, the recent low-budget active learning technique ProbCover is highly sensitive to its difficult-to-tune radius value. On the other hand, GMM is extremely straightforward and easy to apply, with high-quality implementations widely available.
>
> We would like to also emphasize that despite its simplicity and strength, GMMs have not been broadly used in the active learning community of late (including outside the very-low-budget regime): many papers use $k$-means variants, but our experiments show that GMMs often outperform $k$-means in these settings (please refer to the response to the fourth question of Reviewer r9Lv for details). Separately from the choice of clustering method, it also seems that most work “defaults” to hybrid methods (as did e.g. Al-Shedivat et al.) without further considering the tradeoffs. We believe that hybrid methods are indeed likely the best choice in medium- or high-budget settings, but as we demonstrate here, in the very-low-budget setting it seems that very simple selection criteria substantially beat other approaches. We believe these observations can bring new insights to the active learning community.
>
>
> > The paper can benefit from a discussion of practical implications of being able to be more label-efficient at meta-test time grounded in a real-world example.
>
> This is indeed a very good presentation suggestion. There are several natural examples where label efficiency at meta-test time is of paramount important, such as:
>
> - Few-shot learning for medical imaging: when a meta-learner is deployed for new imaging instruments or modalities, such as switching between differently-configured CT scanners or from CT scans to ultrasound images, the resulting images can change dramatically. These changes in features are hard to predict, and because reliable annotations of medical images are extremely expensive, the meta-learner should adapt to the new modality very quickly. By strategically choosing context samples in the new modality, the meta-learner can adapt with only a few samples, and effectively utilize new types of medical imaging data.
> - Industrial automation: consider a factory which frequently switches between manufacturing different types of goods. A quality control system should learn to adapt to the new setting without extensive human labeling for what to look for.
>
> We will add one of these examples throughout the main body, so that readers can better understand implications of the setting and our method.

---

> ### Author Response · Authors · 2023-11-20
> **Response to Reviewer PTqo (2/2)**
>
> > Theoretical justification is very basic and presumed to hold in meta-learning case with ad-hoc justification.
>
> It is true that our theorem is of limited scope. We believe it is useful as an illustrative example for “why” the method can be a good idea, rather than presuming that the exact mechanism elucidated in the theorem applies for practical meta-learning regimes: indeed, Figure 5 shows a simple setting where the theorem fails. (Almost no theorems in machine learning theory precisely apply to practical deep learning settings.) We do believe, however, that the reasoning described in Proposition 1/Figure 4 and in the discussion around Figure 5 provide good motivation for the method.
>
>
>
> > For Tables 1, 2, 3, how many runs are used to compute error bars?
>
> As specified in the last sentence of the second paragraph of Section 4.1, we evaluated a single meta-learned algorithm on 600 meta-test tasks, and reported $95%$ confidence intervals across those tasks. This follows the widely-used convention in the meta-learning literature, including MAML, ANIL, SimpleShot, and more.
>
>
>
> > Is there a hypothesis for why GMM helps at test time but not at meta-train time? This is an important discrepancy that is worth understanding deeper.
>
> We do agree that a deeper understanding of this problem is important. We have a hypothesis as to why this might be the case.
>
> The GMM-based method provides context sets which give better-performing predictors. Doing this at meta-training time, however, makes a given task easier. With optimal context set selection, even mediocre features might yield a good meta-learned model on that task; the problem becomes “too easy,” and the meta-learned features will not give such clear classifications on harder problems. With worse selection of context sets, however, the meta-learner must learn better features, such that even mediocre labelings still achieve reasonable task performance.
>
> We conjecture the improvement of Al-Shedivat et al. in meta-training time might be partially explained by this hypothesis. Recall that Al-Shedivat et al. create clusters of unlabeled samples using k-means++, then select the points from each cluster with the highest entropy. As discussed previously, this will tend to select points near cluster boundaries; the meta-learner should then find features that separate clusters very well, so that these points remain informative enough for the final classifier.
>
> One might be curious about the combination of Al-Shedivat et al. and our GMM method, since their method is for meta-training time and ours is for meta-test time. We thus trained a ProtoNet on MiniImageNet using Al-Shedivat et al., denoted as [1], rather than the Random selection as in the experiments in the paper. We then employed Random, [1], and GMM methods at meta-testing time.
> As shown in the table below, models meta-trained with [1] perform somewhat better with [1]-based meta-testing than random meta-testing. When the model is meta-trained with random selection, the two meta-testing methods are about the same. For both meta-training methods, however, meta-testing with the GMM method substantially outperforms the other two approaches.
>
> Usually ProtoNet is trained for 5,000 iterations; these results use only 1,000 or 2,000 iterations, since meta-training with [1] requires substantial computational overhead (mainly, running k-means++ for every task considered during training).
>
>
> | Train $\rightarrow$ Test | [1] $\rightarrow$ Random | [1] $\rightarrow$ [1]    | [1] $\rightarrow$ GMM    | Random $\rightarrow$ Random | Random $\rightarrow$ [1]  | Random $\rightarrow$ GMM |
> | ------------ | ------------ | ------------ | ------------ | --------------- | ------------- | ------------ |
> | 1000 iter    | 33.40 $\pm$ 0.22 | 34.76 $\pm$ 0.24 | 40.65 $\pm$ 0.28 | 34.94 $\pm$ 0.24   | 33.64 $\pm$ 0.25 | 38.94 $\pm$ 0.26 |
> | 2000 iter    | 36.17 $\pm$ 0.25 | 36.87 $\pm$ 0.25 | 40.76 $\pm$ 0.28 | 35.43 $\pm$ 0.26   | 36.30 $\pm$ 0.2)  | 42.49 $\pm$ 0.28 |
>
> **Table. MiniImageNet using ProtoNet trained with [1] or Random selection**
>
>
> [1] M. Al-Shedivat, L. Li, E. Xing and A. Talwalkar, “On data efficiency of meta-learning”,  AISTATS 2021.

---

> ### Author Response · Authors · 2023-11-22
> **A Reminder to Reviewer PTqo**
>
> Dear Reviewer PTqo,
>
> We kindly remind Reviewer PTqo that the discussion period will end in approximately 12 hours. We highly appreciate the reviewer for valuing our work and providing constructive feedback.
>
> In response to  the reviewer’s valuable comments, both in the common response and directly to the reviewer, we have clarified questions, provided real world examples as suggested, and conducted more experiments, specifically on both hybrid active learning methods and the suggested AL-Shedivat et al.
>
> Your feedback on our rebuttal and revised paper would be greatly appreciated. We are willing to address any remaining doubts or questions. Thank you for your time and consideration.

---

> > ### Comment · Reviewer_PTqo · 2023-11-23
> > **Post author response**
> >
> > Thank you for responding to my questions and the weaknesses I raised. I will maintain my score of 6 since I believe my assessment when writing the review still holds; namely this is a strong empirical paper with limited technical novelty.

---

### Official Review · Reviewer_xa6o · 2023-11-06

**Soundness:** 2 fair
**Presentation:** 2 fair
**Contribution:** 2 fair
**Rating:** 5
**Confidence:** 3

**Summary:**

This paper introduces an approach that integrates active learning into meta-learning with the goal of enhancing data efficiency when selecting context points during the meta-testing phase. The paper starts with an analysis of where in the meta-learning process active learning can be applied, along with a concise review of meta-learning and active learning methods. Empirical evidence is presented to demonstrate that actively selecting context points during meta-training does not significantly impact meta-learning but does prove beneficial during meta-testing. In this context, the authors propose a Gaussian Mixture Model-based acquisition function, which stands as the primary technical contribution of this paper. In essence, this method employs meta-trained features to model a mixture of k Gaussian distributions, with k equating to the label budget, often referred to as batch size in active learning. The empirical results, based on experiments conducted across four few-shot image datasets, indicate that the proposed GMM-based acquisition strategy outperforms other acquisition strategies when integrated into meta-testing.

**Strengths:**

Originality: While the concept of integrating active learning into meta-learning is interesting, it's worth noting that this idea, at a high level, has been explored before. However, the authors' empirical findings regarding the placement of active learning align with existing research, indicating that actively selecting context points during meta-training does not significantly improve few-shot performance. The use of an acquisition function based on a mixture of Gaussians has demonstrated effective performance across various few-shot classification tasks.

Clarity: The paper effectively communicates its main ideas. The explanation of how active learning can be incorporated into meta-learning is well-presented and offers valuable guidance for practical applications.

Significance: The proposed method, utilizing a GMM-based acquisition strategy, has exhibited promising results when compared to several acquisition strategies employed in active learning. This achievement is particularly noteworthy across multiple few-shot image classification and vision regression datasets.

**Weaknesses:**

The primary technical contribution of this paper lies in the GMM-based acquisition function, which is suggested to outperform other acquisition functions in scenarios with extremely limited annotation budgets, as required by meta-learning. However, the results presented in Figure 2 do not convincingly demonstrate a substantial performance improvement of the proposed GMM-Based method over Typiclust.

Furthermore, selecting the samples close to the cluster centre can better capture the diversity. However, it's important to note that several hybrid active learning approaches already consider both uncertainty and diversity. For example, BEMPS [1] integrates these aspects. To better highlight the advantages of the proposed methods, a more comprehensive examination within the context of active learning is advisable.

References
*  W.Tan, L.Du, and W.Buntine, “Diversity enhanced active learning with strictly proper scoring rules,” in Advances in Neural Information Processing Systems, 2021,pp.10906– 10918.

**Questions:**

The reviewer has a question about the setup of active learning in meta-testing.
Was the acquisition just run to acquire N*K samples?  Or was the acquisition run multiple times until the total annotation budget N*K was exhausted? If the later case, was the meta-trained model retrained after each acquisition iteration? And what was the batch size (No. Samples acquired) used in each iteration?

---

> ### Author Response · Authors · 2023-11-20
> **Response to Reviewer xa6o (1/2)**
>
> Thank you for your comments. Please see the common response above, in addition to discussion here.
>
>
> > The primary technical contribution of this paper lies in the GMM-based acquisition function, which is suggested to outperform other acquisition functions in scenarios with extremely limited annotation budgets, as required by meta-learning. However, the results presented in Figure 2 do not convincingly demonstrate a substantial performance improvement of the proposed GMM-Based method over Typiclust.
>
> We would like to emphasize that our goal in Figure 2 was not to claim that the GMM-based method significantly outperforms other methods such as Typiclust; indeed, we described the results as saying that it “matches or outperforms other low-budget methods.” Figure 2 is a motivational experiment for active meta-learning, justifying that the GMM method is a reasonable active learning scheme in the very-low-budget but non-meta setting. In our main experiments for active meta-learning, in Section 4, we believe we do convincingly demonstrate a substantial performance improvement over Typiclust: for instance, the first results in Tables 1 and 2 show improvements over Typiclust varying between two and eight percentage points of accuracy.
>
>
>
> > Furthermore, selecting the samples close to the cluster centre can better capture the diversity. However, it's important to note that several hybrid active learning approaches already consider both uncertainty and diversity. For example, BEMPS [1] integrates these aspects. To better highlight the advantages of the proposed methods, a more comprehensive examination within the context of active learning is advisable.
>
> Thank you for this important suggestion. We added an experiment comparing our approach with hybrid active learning methods, with results listed in the common response. These results show that the proposed GMM-based method significantly outperforms hybrid methods, which we think was somewhat expected since uncertainty-based methods perform so poorly (often much worse than random selection) in these settings.
>
> Most uncertainty measures should be high near decision boundaries and/or outlier points; this is undesirable for low-budget settings, since they are hard to generalize from in “initial” stages of learning. Methods like our GMM-based proposal ensure that the context set points are “near” most of the target set, helping in the early phases of learning which are all that meta-learning considers. Hybrid methods considering both uncertainty and diversity are likely to help in middle-budget regimes, but not in extremely-low budget settings like meta-learning. We’ve added discussion of this point to Appendix L of the revised paper.
>
> For BEMPS in particular, as we mentioned in the common response, it is hard to apply to typical meta-learning settings, since it requires a posterior over models. Bayesian active learning papers mostly compare only to other Bayesian active learning methods. We have added a citation to Section 3.1, however.

---

> ### Author Response · Authors · 2023-11-20
> **Response to Reviewer xa6o (2/2)**
>
> > Was the acquisition just run to acquire NK samples? Or was the acquisition run multiple times until the total annotation budget NK was exhausted? If the later case, was the meta-trained model retrained after each acquisition iteration? And what was the batch size (No. Samples acquired) used in each iteration?
>
> Thanks for raising the question. As briefly mentioned in Section 3.1, we mostly considered the former scenario where we sample $N\times K$ samples at once. Although iterative sampling is more common in active learning, we focused on this scenario for the following two reasons:
>
>
> 1. Even when we iteratively label additional samples, for most meta-learning algorithms (other than MAML), this does not change the features we will use for our meta-learning methods. Even for other optimization-based methods such as ANIL, since the feature extractor is not updated during adaptation on a context set, the features stay the same for iterative process of active learning. As we demonstrated with ProtoNet in Figure 9 (c)-(d) in Appendix K (details about experiments in [1] below), when we iteratively add more labeled samples, the performance does not change much as the features do not change. In this case, selecting $N \times K$ samples at once is not very different from the iterative process, but it is computationally cheaper.
> 2. If we iteratively add labeled samples, we will quickly move beyond the few-shot regime of meta-learning, which is often not very practical in real world settings. Suppose we have a meta learner trained in $5$-way $1$-Shot. It is reasonable to add $5$ samples per iteration, since that is the minimum number to cover all the classes. But only after $5$ iterations, it will reach few-shot regime where we typically have $25$ labeled context samples. This is even less practical for $5$-Shot case.
>
> We added the above explanation and experiments in Appendix K. If this is still not clear, we are eager to elaborate it further.
>
> [1] We used MAML and ProtoNet trained on MiniImageNet in $5$-way $1$-Shot with randomly sampled context and target data. In meta-test time, we label $K$ (the number of shots) samples using various active learning methods at each iteration until the total number of context samples reaches $50$. Again, for each iteration, we evaluated on $600$ test tasks.

---

> ### Author Response · Authors · 2023-11-22
> **A Reminder to Reviewer xa6o**
>
> Dear Reviewer xa6o,
>
> We kindly remind Reviewer xa6o that the discussion period will end in approximately 12 hours. In response to your valuable feedback, we have addressed two points of misunderstanding and incorporated additional experiments, particularly in comparison to hybrid active learning methods as the reviewer suggested (see the common response above). We would greatly appreciate your feedback on any remaining unclear points in our responses and revisions. We are willing to address any doubts or questions until the last minutes. Thank you for your time and consideration.

---

> > ### Comment · Reviewer_xa6o · 2023-11-23
> >
> > I appreciate that the authors took to respond to my questions and concerns.
> >
> > In regard to GMM, if its performance is only comparable with those competitors considered in Figure 2, the contribution of the proposed GMM to low-budget active learning is questionable.

---

> ### Author Response · Authors · 2023-11-23
> **Response to the Reviewer xa6o**
>
> Thank you for your feedback. However, we want to **re-emphasize** that we do not claim that our contribution is for improving low-budget active learning as we have already addressed in the rebuttal.
>
> The setting for Figure 2 is "regular" supervised image classification tasks for low budget whereas our target task is meta-learning (either classification or regression). **We want to emphasize that they are two very different tasks.** As we have answered in the response, we provided Figure 2 as a motivational experiment for active meta-learning, justifying that the GMM method is a reasonable active learning scheme in the very-low-budget but non-meta-learning setting.
>
> As the reviewer stated the performance of GMM is comparable (sometimes better) with other competitors for image classification, thus there is no our contribution. On the other hand, for meta-learning, as our extensive experiments show, the proposed GMM significantly outperforms other active learning methods.
>
> **Once again, we earnestly request the reviewer's understanding of the distinctions between them.**
>
> If this was the only remaining concern, we would greatly appreciate the reviewer's re-evaluation of our work based on clarification.
> Thank you for your time and consideration.

---

### Author Response · Authors · 2023-11-20
**Common Response (3/3)**

> Comparison with diversity and uncertainty hybrid methods

Both Reviewer xa6o and PTqo raised concerns about the effectiveness of the proposed GMM-based method compared to existing hybrid active learning methods such as BEMPS [1] and Al-Shedivat et al. [2] that utilize both uncertainty and diversity measures to select new data points to annotate.

We did not include the comparison with hybrid methods because 1) we believed that there has been consensus that diversity measure is significantly more important than uncertainty for low budget active learning through recent works on low budget active learning, and 2) our experiments consistently show that uncertainty-based methods are often significantly worse than even random selection. Based on the reviewers’ suggestions, though, it does make sense to ensure the combination doesn’t fix those performance issues. We thus compared the GMM method with Weighted Entropy, BADGE, and [2], on MiniImageNet (with MAML) and FC100 (with ProtoNet), in the table below.


- Weighted Entropy: [3] proposed weighted expected error active learning method for binary classification. We generalized their approach to multi-class classification, in which case it becomes weighted entropy, with weights based on likelihood computed using soft k-means.
- BADGE [4]: a popular hybrid active learning method. It uses $k$-means++ initialization, with embeddings derived from the gradients of loss with respect to the output of the penultimate layer, using pseudo labels.
- Al-Shedivat et al. [2]: it first clusters unlabeled samples using k-means++, and selects samples per cluster using entropy. They propose this method for meta-training time, but we apply it here in meta-test time. For a detailed analysis for [2] at meta-training time, please refer to the response to the last question of Reviewer PTqo.

We did not include BEMPS in this experiment, because it is a Bayesian active learning method which can only be applied in settings where we can provide a posterior distribution over model parameters, such as ensemble methods or MC Dropout. Common meta-learning methods including the ones we benchmarked do not provide these posteriors, and it is impractical to train multiple meta learners just for the purpose of active learning. It is rare for non-Bayesian active learning papers to compare to Bayesian methods, or for Bayesian papers to compare to non-Bayesian methods (including the BEMPS paper), because of this underlying large difference in model setting.


| $Pick^{eval}_\theta$ | 1-Shot train strat. | 1-Shot unstrat.  | 5-Shot train strat. | 5-Shot unstrat.  |
| -------------------- | ------------------- | ---------------- | ------------------- | ---------------- |
| Weighted Ent.        | 22.69 $\pm$ 0.18    | 32.27 $\pm$ 0.32 | 23.75 $\pm$ 0.25    | 46.80 $\pm$ 0.33 |
| BADGE                | 27.71 $\pm$ 0.18    | 34.30 $\pm$ 0.21 | 41.37 $\pm$ 0.28    | 58.79 $\pm$ 0.24 |
| Al-Shedivat et al. [2]                  | 30.59 $\pm$ 0.28    | 33.73 $\pm$ 0.24 | 38.24 $\pm$ 0.29    | 54.87 $\pm$ 0.26 |
| GMM (Ours)           | 33.34 $\pm$ 0.24    | 37.68 $\pm$ 0.19 | 54.35 $\pm$ 0.20    | 59.05 $\pm$ 0.20 |
**Table. MiniImageNet using MAML**


| $Pick^{eval}_\theta$ | 1-Shot train strat. | 1-Shot unstrat.  | 5-Shot train strat. | 5-Shot unstrat.  |
| -------------------- | ------------------- | ---------------- | ------------------- | ---------------- |
| Weighted Ent.        | 31.80 $\pm$ 0.20    | 28.94 $\pm$ 0.19 | 40.40 $\pm$ 0.25    | 39.95 $\pm$ 0.25 |
| BADGE                | 30.91 $\pm$ 0.23    | 29.29 $\pm$ 0.28 | 43.85 $\pm$ 0.22    | 44.00 $\pm$ 0.29 |
| Al-Shedivat et al. [2]                  | 30.93 $\pm$ 0.22    | 30.43 $\pm$ 0.24 | 41.76 $\pm$ 0.27    | 43.41 $\pm$ 0.29 |
| GMM (Ours)           | 34.23 $\pm$ 0.23    | 35.03 $\pm$ 0.23 | 46.30 $\pm$ 0.21    | 47.03 $\pm$ 0.20 |
**Table. FC100 using ProtoNet**

We can see that in both settings, the GMM method outperforms all hybrid methods, often by a large margin. As we shall see in the experiment where we compare GMM with the variants of $k$-means, hybrid methods do not perform as well as diversity-based methods like $k$-means (please refer to fourth question raised by Reviewer r9Lv). This experiment shows that in this very-low-budget regime, diversity measures are significantly more important than uncertainty measures; we believe this is not yet commonly known in the community, given that multiple reviewers were not sure of this. We included this result in Appendix L along with Table 14.

[1] W. Tan, L.Du, W.Buntine, “Diversity enhanced active learning with strictly proper scoring rules”, NeurIPS 2021.

[2] M. Al-Shedivat, L. Li, E. Xing, A. Talwalkar, “On data efficiency of meta-learning”,  AISTATS 2021.

[3] H. Nguyen, A. Smeulders, “Active Learning Using Pre-clustering”, ICML 2004.

[4] J. Ash, C. Zhang, A. Krishnamurthy, J. Langford, A. Agarwal, “Deep Batch Active Learning by Diverse, Uncertain Gradient Lower Bounds”, ICLR 2020

---

### Author Response · Authors · 2023-11-20
**Common Response (2/3)**

> The proposed GMM method is not tailored for meta learning and technical novelty is limited.

Although we do briefly note in the paper that the GMM method can be applied to active learning settings outside of meta-learning, we would first like to argue that it is indeed particularly suited to meta-learning:


1. In recent years, active learning community has mainly focused on batch selection in high-budget settings, since deep learning requires a large amount of data. Although some recent works such as Typiclust attempted to address active learning in low-budget settings, their budgets (around 60, 600, or 6000) are still much larger than typical budgets for meta-learning (for 5-way problems, the budget is 5 for 1-shot or 25 for 5-Shot). As shown in Figure 2, the proposed GMM method matches or outperforms other low-budget active learning methods in this very-low-budget regime, not considered by previous work.
2. Providing good “reference points” in the context set is important in meta-learning. When selecting context samples based on maximum uncertainty (as measured by, say, entropy), we will tend to select points near decision boundaries and in less dense regions. This is not desirable for the very-low-budget regime of meta-learning, where we need to focus on labels for the most-typical points. Previous low-budget methods look for points in locally dense regions, not global coverage; see our response to Reviewer r9Lv’s last question for more. On the other hand, the GMM method ensures that the context set is not too far away from the target set, even in the presence of a few outliers.


So far, active learning methods have covered the following budget regimes:

- low-budget: low budget active learning (mainly considering diversity)
- mid-budget: hybrid active learning (uncertainty + diversity)
- high-budget: uncertainty-based active learning

As shown in our paper, existing low-budget and uncertainty-based methods prove ineffective in the very-low-budget settings of meta-learning. New experiments in our response below also show that suggested hybrid methods from various reviewers do not perform as well as our GMM-based method, either.

We also provide our contributions below. We hope the reviewers consider the implication and meaning of our work, rather than the complexity of our proposed algorithm.


1. We clarify that “active meta-learning” can mean several things. Most previous work has used active selection of tasks at meta-training time. We thus explore active selection of context sets, which is far less explored previously, and point out that this active context set selection can occur in several places (Sec 2). We decide to focus on active selection of context sets at eval time, as it has not yet been thoroughly studied. We also identify a major challenge in the traditional setup for meta-learning in classification (Sec 2.1) related to stratification of samples, which has not been previously considered.
2. Lacking much previous work in active meta learning for context selection, it is natural to apply existing active learning algorithms (Sec 3.1). However, uncertainty-based methods generally perform worse than random selection (Table 1-4, 6-7). Diversity-based methods (particularly for low budget setting) do not perform significantly better than random selection in many cases, either (Table 2-7); this is counter-intuitive because context selection in meta-test time should be in the regime of low-budget active learning.
3. We propose to choose data points closest to the cluster means from a GMM (Sec 3.3), using meta-learning specific features from the penultimate layer (as justified in Sec 3.2 and Appendix I.) For metric-based meta-learning methods like ProtoNet, GMM has a similar justification to other representation-based methods. For optimization-based methods, however, the GMM method has additional justification: in low-budget regimes, many models trained on the selected data become max-margin classifiers. When the data distribution in the meta-learned features follows a class-conditional Gaussian with orthonormal means and spherical variance (a strong but not totally implausible assumption), we prove in Proposition 1 that our strategy yields a Bayes-optimal classifier. For more general settings, we argue that GMM is still a good method based on being an efficient set cover (Figure 4).
4. We thoroughly demonstrated that the simple GMM method significantly outperforms more complicated active learning methods for few-shot image classification (Sec 4.1 and Appendix G), cross-domain classification (Sec 4.2), and meta learning regression tasks (Sec 4.3) with all types of meta-learning algorithms: metric-based (Table 1), optimization-based (Table 2), model-based (Table 5), and pre-training-based (Tables 3-4).
5. We also validated that active selection of context sets in meta-training does not improve a meta-learner over random sampling (Appendix J), verifying observations from previous works.

---

### Author Response · Authors · 2023-11-20
**Common Response (1/3)**

We appreciate all of the reviewers’ valuable feedback. In this common response, we address the concerns raised by multiple reviewers. Other responses are provided to individual reviewers. All the changes are reflected in the revised version, in blue font. If the reviewers have any further concerns or questions, we are eager to continue discussion.

---

### Author Response · Authors · 2023-11-22
**Discussion Ends in about a Day**

Dear Reviewers,

We again deeply appreciate your constructive feedback. We have carefully addressed all your comments and suggestions, making necessary revisions to our paper. With the discussion deadline approaching in about a day, we kindly remind the reviewers to consider our responses and re-evaluate our work. We are available for any additional questions or concerns, eager to engage in further discussion. Your feedback is invaluable to improving the quality of our work. We look forward to hearing from you soon. Thank you.

---

> ### Comment · Area_Chair_pVdn · 2023-11-22
>
> Dear reviewers,
>
> This is an additional reminder:
>
> The authors have posted responses and answers to questions raised by the reviews. Please take a look and update you reviews and rating accordingly. Even if your decision remains the same, please update your review to indicate you have read the author response and why your decision is unchanged.
>
> Thank you for your critical contributions to this conference.

---

### Meta-Review · Area_Chair_pVdn · 2023-12-05

**Metareview:**

This submission investigates the use of active learning in meta-learning, specifically focusing on the meta-testing phase of learning. Here the authors recognize the need for an active learning algorithm that can be effective in the ultra-low data regime (<25 examples) and identify and  validate the use of a Gaussian Mixture Model (GMM) based sampling strategy.

The main salient weaknesses identified by reviewers include: limited theoretical justification (although some limited theory is included) and the lack of technical novelty needed to solve the problem.

Weighing this against the strengths of the paper:
- Convincing empirical results (including additional results provided during the discussion)
- Despite the approach being simple, it still needed to be identified and evaluated among several competitors in the non-traditional ultra-low data regime active learning setting.
- Intuition (and additional arguments provided during discussion) around why the GMM based method perform better in this regime and why meta-test time is more amenable to this approach that meta-train time.

Edit by the PCs: Based on the recommendation by the SAC, we recommend to reject this paper.

**Justification For Why Not Higher Score:**

For a higher score I would expect a more in-depth theoretical (and/or empirical) analysis to provide a stronger understanding of "why" the proposed method is working as well as it is.  Currently, the authors provide strong empirical evidence that the method does work and some limited theoretical arguments and informal intuitions/hypotheses as to why it works.

**Justification For Why Not Lower Score:**

The submission attacks a practically important and relatively under-explored problem and provides what seems to be an practical and effective solution. The authors have also responded to all, in my opinion, reasonable concerns raised by reviewers.

---

### Decision · Program_Chairs · 2024-01-16

Reject